



# Information content and aerosol property retrieval potential for different types of in situ polar nephelometer data

Alireza Moallemi[1], Rob L. Modini[1], Tatyana Lapyonok[2], Anton Lopatin[3], David Fuertes[3], Oleg Dubovik[2], Philippe Giaccari[4], Martin Gysel-Beer[1]

[1]Laboratory of Atmospheric Chemistry, Paul Scherrer Institute, Villigen PSI, 5232, Switzerland
[2]Univ. Lille, CNRS, UMR 8518 - LOA - Laboratoire d'Optique Atmosphérique, F-59000 Lille, France
[3]GRASP-SAS, Remote sensing developments, Université de Lille, 59655 Villeneuve d'Ascq Cedex, France
[4]Micos Engineering GmbH, Dübendorf, CH-8600, Switzerland

*Correspondence to*: Rob L. Modini (robin.modini@psi.ch)

**Abstract.**

Polar nephelometers are in situ instruments used to measure the angular distribution of light scattered by aerosol particles. These type of measurements contain substantial information about the properties of the aerosol being probed (e.g. concentrations, sizes, refractive indices, shape parameters), which can be retrieved through inversion algorithms. The aerosol property retrieval potential (i.e., information content) of a given set of measurements depends on the spectral, polarimetric and

angular characteristics of the polar nephelometer that was used to acquire it. To explore this issue quantitatively, we applied Bayesian information content analysis and calculated the metric *Degrees of Freedom for Signal* (*DOFS*) for a range of simulated polar nephelometer instrument configurations, aerosol models and test cases, and assumed levels of prior knowledge about the variances of specific aerosol properties. Assuming a low level of prior knowledge consistent with an unconstrained ambient/field measurement setting, we demonstrate that even very basic polar nephelometers (single wavelength, no

polarization capability) will provide informative measurements with very high retrieval potential for the size distribution and refractive index state parameters describing simple unimodal, spherical test aerosols. As expected, assuming a higher level of prior knowledge consistent with well constrained laboratory applications leads to a reduction in potential for information gain via performing the polarimetric measurement. This analysis allows us to better assess the impact of different polar nephelometer instrument design features in a consistent manner for retrieved aerosol parameters. The results indicate that the

addition of multi-wavelength and/or polarimetric measurement capabilities always leads to an increase in information content, although in some cases the increase is negligible: e.g. when adding a fourth, near-IR measurement wavelength for the retrieval of unimodal size distribution parameters, or if the added polarization component has high measurement uncertainty. By considering a more complex bimodal, non-spherical aerosol model, we demonstrate that performing the more comprehensive spectral and/or polarimetric measurements leads to very large benefits in terms of the achieved information content. We also

investigated the impact of angular truncation (i.e., the loss of measurement information at certain scattering angles) on information content. Truncation at extreme angles (i.e., in the near-forward or –backward directions) results in substantial decreases in information content for coarse aerosol test cases. However for fine aerosol test cases, the sensitivity of *DOFS* to



extreme angle truncation is noticeably smaller and can be further reduced by performing more comprehensive measurements. Side-angle truncation has very little effect on information content for both the fine and coarse test cases. Furthermore, we

demonstrate that increasing the number of angular measurements generally increases the information content. However, above a certain number of angular measurements (~20-40) the observed increases in *DOFS* plateau out. Finally, we demonstrate that the specific placement of angular measurements within a nephelometer can have a large impact on information content. As a proof-of-concept, we show that a reductive greedy algorithm based on the *DOFS* metric can be used to find optimal angular configurations for given target aerosols and applications.

## 1 Introduction

Aerosols are condensed phase particles suspended in the air that are distributed ubiquitously in the atmosphere. Aerosol particles have a broad range of sizes spanning orders of magnitudes, and they possess diverse physical and chemical properties. They affect the global climate either by directly scattering or absorbing solar radiation, or by influencing cloud formation processes (Boucher et al., 2013). Aerosol particles are also one of the major components of air pollution and aerosol exposure

has been linked to cardiovascular and pulmonary diseases and premature deaths (Cohen et al., 2017; Lelieveld et al., 2015).

Due to the importance of aerosol particles for the global atmospheric system and public health, a multitude of methods have been developed to measure and characterize these particles with both in situ and remote sensing instruments. Remote sensing methods rely on the interaction of aerosols with solar radiation or laser light, and typically involve detection of the elastically

scattered light. In particular, many instruments have been designed to measure the angular dependence of scattered radiance (radiometry), with optional measurement of its dependence on polarization state (polarimetry). Radiometry and polarimetry are the cornerstones of ground-based and space-borne remote sensing applications because the resulting measurements contain retrievable information on aerosol microphysical properties, which can be obtained via inversion algorithms. Well-known examples of space-borne remote sensing instruments include the Moderate Resolution Imaging Spectrometer (MODIS; King

et al., 1992), the Multi-angle Imaging SpectroRadiometer (MISR; Lee et al., 2009), and the Polarization and Directionality of Earth's Reflectance (POLDER; Deuzé et al., 2001) instrument. In terms of ground-based polarimetric observations, the prime example is the AEosol RObotic NETwork (AERONET; Holben et al., 1998), which is a coordinated network of sun photometers at more than 600 sites worldwide.

Three design aspects are fundamentally important for aerosol polarimetry instruments: i) the spectral coverage, i.e. the number of wavelengths at which light scattering is measured, ii) the polarization measurement capability, and iii) the number and position of the probed angles. Over the last few decades, technological advancements have led to substantial instrumentation improvements in terms of all three of these design aspects (Dubovik et al., 2019). Parallel to these instrumentation developments, advanced aerosol property retrieval algorithms have been designed to better utilize these more informative





polarimetric measurements. For example, GRASP-OPEN (Generalized Retrieval of Aerosol and Surface Properties) is a well-established retrieval algorithm that was designed to take advantage of enhanced polarimetric measurements in order to improve the scope and accuracy of aerosol property retrievals (Dubovik et al., 2011, 2014, 2021).

Polar nephelometers, which are in situ instruments for radiometric and polarimetric aerosol measurements, have a rich history
dating back to the 1940s (Waldram, 1945). The main physical quantity measured by polar nephelometers is the phase function (denoted here as *PF*), which is a measure of angular distribution of scattered light radiance by aerosol particles given non-polarized incident light. A subclass of polar nephelometers are also capable of measuring the polarized phase function (denoted here as *PPF*), which is the angular distribution of the portion of linearly polarized scattered light given non-polarized incident light. Note that some polar nephelometers are capable of measuring more scattering quantities (e.g. Hu et al., 2021), but in this
study we only focus on instruments that measure *PF* and *PPF*.

A variety of polar nephelometer designs has been introduced over the years. The first design class of these instruments is the goniometer-type polar nephelometer, which measures angularly-resolved light scattering using a rotating detector (e.g. Waldram, 1945). The major disadvantage of these types of instruments is the long sampling time required to measure a full
phase function, which severely limits their usability in field applications. A second design class is the multi-detector type instrument, which enables more rapid measurements by employing detectors at fixed positions to probe *PF* or *PPF* at discreet angles simultaneously. Multi-detector nephelometers can have either very basic designs (e.g. four angles and only one wavelength; Li et al., 2019), or more complex designs, such as the instruments introduced by Barkey et al. (2007) and Nakagawa et al. (2016), which measure both *PF* and *PPF* at 21 and 17 angles, respectively. A third design class is the laser
imaging nephelometer (Dolgos and Martins, 2014; Ahern et al., 2022). The laser imaging nephelometer illuminates aerosol particles using laser light and collects the scattered light on a charged coupled device (CCD). The angularly resolved scattering measurements are then extracted from the image collected by the CCD. An example of a laser imaging nephelometer is the polarized imaging nephelometer introduced by Dolgos and Martins (2014), which measures *PF* and *PPF* over 174 angles and at three different wavelengths.


The angularly-resolved measurements provided by polar nephelometers enable the retrieval of a large number of aerosol properties via inversion algorithms such as GRASP-OPEN (which can be applied in a versatile manner to both remote sensing and in situ measurement data: Espinosa et al., 2019; Dubovik et al., 2014, 2021). Furthermore, since these instruments provide measurements that are in principle quite similar to remote sensing data, they can be used as well-constrained testbeds for
evaluation and improvement of the inversion algorithms used by the remote sensing community (Schuster et al., 2019). However, the high dimensionality of polar nephelometer measurements with respect to spectral, polarimeteric, and multi-angle measurement options also presents an instrument design challenge. In particular, it raises the questions: i) which types of measurements should be prioritized and optimized in order to maximize the information content concerning the underlying



aerosol properties, and ii) can relatively non-informative measurement types be identified in order to prevent overdesign and complexities that could hamper robustness or cost-effectiveness?

Bayesian information content analysis is a tool that has been used extensively by the remote sensing community to guide such instrument design challenges by enabling the calculation of quantitative information content metrics for specific instrumentation configurations, settings, and applications (Rodgers, 2000). For example, the framework has been used to demonstrate the importance of polarized and multi-angle scattering measurements in the context of space-borne satellite measurements and ground-based sun photometers (Knobelspiesse and Nag, 2018; Chen et al., 2017; Ding et al., 2016; Xu and Wang, 2015; Ottaviani et al., 2013; Knobelspiesse et al., 2012). However, to the best of our knowledge, the framework has not yet been applied to in situ light scattering measurements in order to tackle the design challenges posed above. As well as being used to assess information content, the Bayesian theoretical framework (Rodgers, 2000) can also be used as an inversion method. In this context, the Bayesian approach can be considered as a particular case of the more general retrieval method employed in GRASP-OPEN, as discussed by Dubovik et al., (2021), where it is also noted that Bayesian-based retrievals can be hampered by the requirement of directly specifying a priori constraints for all aerosol state parameters.

In this study, we use Bayesian framework purely as a method for assessing the information content of synthetic measurement data, in order to investigate the challenges associated with polar nephelometer instrument design. Specifically, we assess and compare the aerosol property retrieval potential of different polar nephelometer instrument configurations given different target applications and assumed prior knowledge. Furthermore, within the theoretical framework, we use a case study to demonstrate how a well-known optimization algorithm (a reductive greedy algorithm) can be used to determine the optimal placements of the angular sensors in a polar nephelometer.

The paper is structured in the following manner. In section 2, we provide general information on the Bayesian information content framework, while in Section 3 we introduce the specific methodological aspects used to construct the information content analysis used in this study. In Section 4 we present the results of the analysis, including an overview of the information content of different polar nephelometer instrument designs, an assessment of the value of spectral and polarimetric measurements, investigation of measurement artefacts associated with scattered light truncation, and the results of the angular sensor placement optimization case study. Finally, we summarize the conclusions of the study in Section 5.

## 2 Theory: Information content analysis and degrees of freedom for signal (DOFS)

In scientific and engineering applications it is typically not possible to measure quantities of interest directly. Instead, the values of such quantities must be inferred indirectly from measurements of causally-related intermediary quantities. These types of inference problems are known generically as inverse problems. Inverse problems can be expressed mathematically by



considering a so called state vector, $x$, that contains the N desired parameters to be retrieved (i.e., the state parameters $x_i$), and an observation vector, $y$, which corresponds to a set of M measurements conducted by an instrument. The vectors $x$ and $y$ are assumed to be related to each other through the following equation:

$$y = F(x) + \varepsilon \,, \tag{1}$$

where F($x$) is referred to as the forward model. The forward model maps the state vector into measurement space based on the underlying physical processes that are involved. For example, the Mie solution to Maxwell's equations is a forward model that relates the microphysical properties of spherical particles $x$ (e.g., size, refractive index) to measurable optical parameters $y$ (e.g., *PF* and *PPF*). The $\varepsilon$ vector in Eq. 1 represents the error originating from measurement noise as well as the error resulting from the limitations of a particular forward model.


The main objective in an inverse problem is to retrieve a best estimate of the state vector ($\hat{x}$) given a measurement vector $y$ and the relationship in Eq. 1. This can be achieved in a number of different ways. In inverse problems related to atmospheric observations, some prior knowledge of the state parameters to be retrieved is typically available. In such cases, Bayesian inference (Rodgers, 2000) is a well-established method for the retrieval of state vectors. For a forward model that describes

the underlying physics of the problem and that can be locally linearized in the state space, a measurement vector $y$ corresponding to a reference state vector, $x_0$, can be written as:

$$y = Kx_0 + \varepsilon \,, \tag{2}$$

where $K$ is the Jacobin matrix of the forward model at the reference point $x_0$ (i.e., $K = \frac{\partial F(x)}{\partial x}\Big|_{x=x_0}$), which has dimensions of [M×N]. Under the assumptions of normally distributed measurement errors and a priori state parameter ranges, Bayes theorem

can be applied to infer an estimated state vector, $\hat{x}$, from the measurement vector $y$. Using Bayes theorem, $\hat{x}$ and its corresponding error covariance matrix $\hat{S}$ then become:

$$\hat{x} = \hat{S}K^T S_\varepsilon^{-1}(Kx_0 + \varepsilon) + \left(I - \hat{S}K^T S_\varepsilon^{-1}K\right)x_a \,, \tag{3}$$

$$\hat{S}^{-1} = K^T S_\varepsilon^{-1}K + S_a^{-1} \,, \tag{4}$$

Detailed descriptions on how to derive these formulae is provided in Chapter 2 of Rodgers (2000). The matrix $S_\varepsilon$ is the

measurement error covariance matrix. The diagonal elements of $S_\varepsilon$ represent the measurement uncertainties for each of the measured elements $y_i$, while the off-diagonal elements describe the covariance between the errors for different elements. The vector $x_a$ is the a priori vector in the state space, which contains what one believes to be the expected values for the state vector prior to the measurement being performed (i.e., the prior knowledge of the state space variables). The matrix $S_a$ is the a priori covariance matrix which describe the expected spread of the state vector parameters in the state space. The matrix $\hat{S}$ is the

retrieval (posterior) error covariance matrix, which describes the spread of the estimated state vector $\hat{x}$. The square root of





diagonal elements of $\hat{\mathbf{S}}$ correspond to the retrieval (posterior) uncertainties of individual state parameters and in this study they are denoted as $\sigma_{post,i}$ for the given parameters $x_i$.

By defining $\hat{\mathbf{S}}\mathbf{K}^{\mathrm{T}}\mathbf{S}_{\varepsilon}^{-1}\mathbf{K}$ as the so-called *averaging kernel matrix* **A**, Eq. 3 can be rewritten as:

$$\hat{x} = \mathbf{A}x_0 + (\mathbf{I} - \mathbf{A})x_a + \varepsilon_x, \tag{5}$$

where **I** is the identity matrix and $\varepsilon_x$ is the retrieval error, which corresponds to the transformation of measurement error from the measurement space to the state space. The averaging kernel matrix **A** captures the sensitivity of the retrieved state vector $\hat{x}$, to the reference state vector $x_0$. The diagonal elements of **A** ($A_{ii}$) are referred to as the *degrees of freedom for signal* ($DOFS_i$) for the given state parameters $x_i$. These elements can take values between 0 and 1. As shown in Eq. 5, the $DOFS_i$ values can be interpreted as the normalized factors used to compute the optimal estimate retrieval result (and retrieval error) as a weighted

average of the transformed measurement errors and the a priori parameter ranges following Bayes theorem. That is, the $DOFS_i$ are quantitative metrics that describe the information content of a given measurement in relation to prior knowledge. For example, a $DOFS_i$ value close to 1 indicates that a particular measurement makes a greater contribution to the retrieval of a given state parameter $\hat{x}_i$ compared to what is known a priori about the value of that state parameter (i.e., $DOFS_i \rightarrow 1$ indicates a more informative measurement). On the other hand, a $DOFS_i$ values closer to zero indicates that a priori knowledge makes

the dominant contribution to the retrieval of a state parameter $\hat{x}_i$, whereas the corresponding measurement only has little influence (i.e. the measurement is non-informative).

To quantify the overall information content of a measurement, the total $DOFS$ for all retrievable state parameters is defined according to the sum:

$$DOFS = \mathrm{trace}(\mathbf{A}) = \sum_{i=1}^{N} A_{ii} \ . \tag{6}$$

Therefore, $DOFS$ assumes values between 0 and N. This means that for a hypothetically perfect measurement $DOFS = N$, the number of state parameters considered in the forward model. Dividing $DOFS$ by N provides the normalized $DOFS$, which we denote as $nDOFS$.

It must be stressed that $DOFS$ values depend intrinsically on the assumed a priori variables and measurement uncertainties when applying Bayesian inference. Therefore, one must be careful not to over-interpret absolute $DOFS$ values without considering their full context. These issues can be partially avoided by using $DOFS$ values in a relative manner, e.g. to rank different designs with respect to achievable information content. As an alternative approach to circumventing the influence of assumed prior knowledge, Alexandrov and Mishchenko (2017) proposed the use of $\mathbf{S}_{\mathbf{pro}}$ as an information content metric,

given by:

$$\mathbf{S}_{\mathbf{pro}} = (\mathbf{K}^{\mathrm{T}}\mathbf{S}_{\epsilon}^{-1}\mathbf{K})^{-1}. \tag{7}$$





Mathematically, $\mathbf{S_{pro}}$ represents the error covariance matrix of a state vector retrieved from a measurement using least squares minimization (LSM) without considering a priori knowledge. The diagonal elements of $\mathbf{S_{pro}}$, denoted as $\sigma_{pro}^2$, represent the variance of the LSM-retrieved state parameters. In this study, we use $\mathbf{S_{pro}}$ as an additional information content metric to support

our discussion of *DOFS* values. A summary of all the information content metrics used in this study is provided in Table 1. Further details about Bayesian interference and *DOFS* can be found in numerous previous studies (Knobelspiesse and Nag, 2018; Ding et al., 2016; Xu and Wang, 2015; Hasekamp and Landgraf, 2005).

**Table 1: Summary of the information content metrics used in this study**

| Parameters | Short description | Formula |
|---|---|---|
| *DOFS$_i$* | DOFS of state parameter *i* | $A_{ii}$ |
| *DOFS* | Total DOFS of all the retrievable state parameters | trace($\mathbf{A}$) |
| *nDOFS* | Total DOFS normalized by the number of retrievable state parameters | DOFS/N |
| $\mathbf{S_{pro}}$ | Propagated measurement error covariance matrix | $(\mathbf{K}^T\mathbf{S}_\epsilon^{-1}\mathbf{K})^{-1}$ |


### 3 Implementation of information content analysis to in situ polar nephelometer data

In this study, the Bayesian framework described in Section 2 was used to investigate the information content and aerosol
property retrieval potential of different polar nephelometer measurement configurations. Figure 1 summarizes the implementation of this analysis, while the subsequent subsections explain inputs, calculation steps and outputs in more detail.



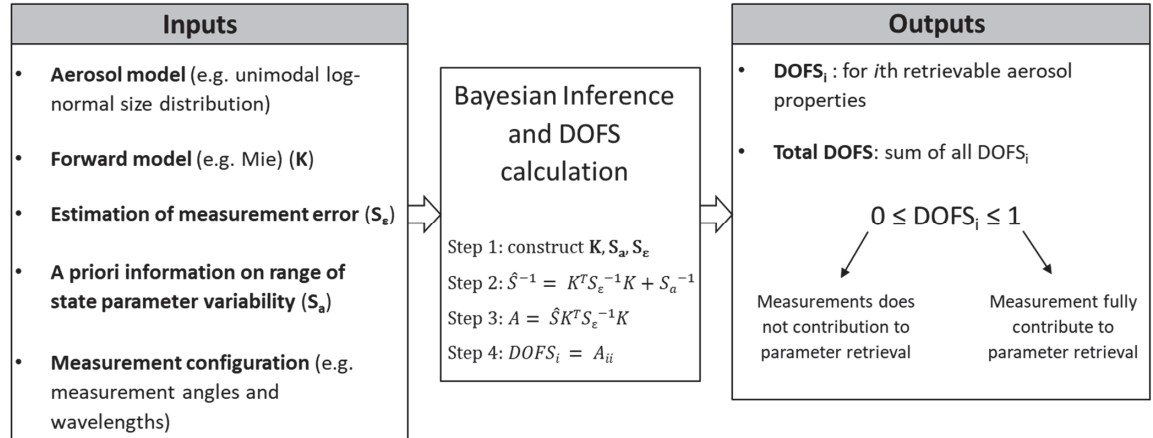

210        **Figure 1. A schematic overview of the implementation of the Bayesian information content analysis**

### 3.1 Measurement configurations

The measurement capabilities of polar nephelometers differ in three main aspects: the spectral characteristics (i.e., the number of measurement wavelengths $N_\lambda$), radiometric only or polarimetric measurement ($PF$ or $PF$ and $PPF$, respectively), and the

angular characteristics (i.e., the number and position of probed angles $N_\theta$). The measurement space vector for a particular instrument will have dimension of either $N_\lambda \times N_\theta$ or $2 \times N_\lambda \times N_\theta$ for radiometric or polarimetric instruments, respectively. That is, we assume that the angular and spectral dimensions are equivalent for $PF$ and $PPF$. In this study, we examined a variety of polar nephelometer configurations that differ according to the above design features.

The general angular geometry of the problem is shown in Fig. 2. The polar scattering angle ($\theta$) range over which discrete light scattering measurements are performed ranges from $\theta = 0°$ (forward direction relative to the direction of the incident light) to $\theta = 180°$ (backward scattered light).





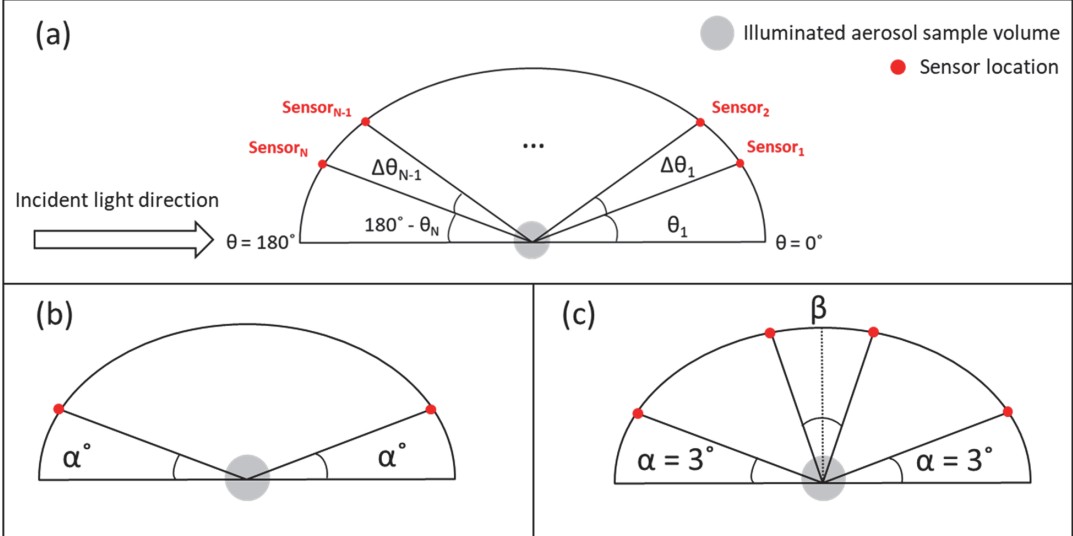


**Figure 2** Schemes of scattering measurement geometry: a) polar angles of sensors, b) truncation around extreme angles, and c) truncation around right angle.

### 3.1.1 Angular characteristics: extreme truncation angles

Probing scattering angles very close to the incident light beam is a general problem in all types of nephelometers (Moosmüller and Arnott, 2003) due to e.g. blockage of this beam, or interference between the incident and scattered light. This leads to truncation of the probed angle range at extreme values representing the forward and backward directions. The difference between the smallest measurable scattering angle $\theta_1$ and 0˚ is referred to as forward truncation angle and is shown as $\alpha$ in Fig. 2b. The difference between the largest measurable scattering angle $\theta_N$ and 180˚ is referred to as backward truncation angle and

is also represented as $\alpha$ in Fig. 2b. That is, we assume that the forward and backward truncation angles are equal in our simulations, since these two angles are similar in many nephelometers (e.g. Bian et al., 2017; Nakagawa et al., 2016; Manfred et al., 2018). To study the effect the extreme angle truncation, we varied $\alpha$ over the set of values [0˚, 1˚, 3˚, 5˚, 10˚, 20˚, 40˚], while keeping $\Delta\theta_i$ constant at 1˚ (a typical value for laser imaging polar nephelometers; Dolgos and Martins, 2014).

### 3.1.2 Angular characteristics: side truncation angles

In some polar nephelometer designs measurements near right angle scattering can also be compromised. For example, in two-beam laser imaging nephelometers (e.g. Dolgos and Martin, 2014) this can occur if an insufficient number of overlapping scattering angles are configured between the forward and backward scattering regions. Here, we refer to this type of





measurement gap as side angle truncation. To investigate side angle truncation we assume that it is symmetric around 90° and represent it by a single angle $\beta$, which represents the angle between the closest sensors to 90° in both the forward and backward

scattering regions as shown in Fig. 2c. We varied $\beta$ over the set of values [1°, 11°, 21°, 31°, 41°, 51°, 61°, 71°, 81°, 91°] while keeping $\Delta\theta = 1°$ and $\alpha = 3°$.

### 3.1.3 Angular characteristics: number of probed angles assuming evenly distributed measurements

For an instrument covering $N_\theta$ distinct angles, each data point has an associated angle $\theta_i$. The angular difference between two adjacent data points, hereafter referred to as sensors, is defined as $\Delta\theta_i := \theta_i - \theta_{i-1}$. As a default setup, we assumed that all sensors

are evenly distributed (i.e., $\Delta\theta_i = \Delta\theta$). To investigate the effect of sensor number on the information content, we varied $\Delta\theta$ over a range that encompasses the sensor numbers of instruments previously reported in the literature. Examples at either end are the instrument described by Li et al. (2019) with four sensors, or laser imaging polar nephelometers with a maximal angular resolution of $\Delta\theta \approx 1°$ (Dolgos and Martins, 2014). The range of simulated $\Delta\theta$ values is shown in Table 2, along with the corresponding $N_\theta$ values for base case truncation angles, i.e., $\alpha = 5°$ and $\beta = 0°$.


**Table 2. Simulated $\Delta\theta$ values and corresponding number of probed angles for a configuration with evenly distributed sensors and $\alpha = 5°$ truncation in forward and backward directions.**

| $\Delta\theta$ [°] | 1 | 2 | 5 | 10 | 17 | 34 | 85 | 170 |
|---|---|---|---|---|---|---|---|---|
| Number of angular measurements ($N_\theta$) | 170 | 85 | 34 | 17 | 10 | 5 | 3 | 2 |

### 3.1.4 Spectral measurement capabilities

We considered three different spectral measurement settings:

1. Single wavelength measurement at 532 nm (1$\lambda$)
2. Three wavelengths measurement at 450 nm, 532 nm, 630 nm (3$\lambda$)
3. Four wavelength measurement at 450 nm, 532 nm, 630 nm, 1020 nm (4$\lambda$)

The selected visible wavelengths, i.e., 450 nm, 532 nm, 630 nm, are chosen to resemble the wavelengths of our custom-built

laser imaging polar nephelometer that is currently being developed. A N-IR wavelength measurement is typically not included in existing polar nephelometer designs. However, measurements at $\lambda = 1020$ nm are performed extensively by ground-based passive remote sensing instruments (e.g. within the AERONET) and they are thought to benefit the retrieval of coarse mode particle properties. Therefore, we used the 4$\lambda$ as a test case to investigate the potential benefits of including a N-IR measurement in an in situ instrument.





### 3.2 Assumptions on measurement error covariance matrix ($\mathbf{S_\varepsilon}$)


The measurement error covariance matrix, $\mathbf{S_\varepsilon}$, (Eq. 3) represents the uncertainty in the measurements of specific parameters (diagonal matrix elements) as well as the co-variance of measurement errors between different data points (non-diagonal elements). For all the analyses we assume normally distributed random measurement errors.

For *PF* and *PPF* at any given wavelength, we chose measurement errors based on those reported by Dolgos and Martins (2014) for their laser imaging polar nephelometer. Specifically, we took the average values of the errors reported by these authors for aerosols with high and low scattering coefficient cases. That is:

$$\sigma_{\mathrm{meas},PF}(\theta_j, \lambda_i) = 0.046 \times PF(\theta_j, \lambda_i), \tag{8}$$

$$\sigma_{\mathrm{meas},PPF}(\theta_j, \lambda_i) = 0.056. \tag{9}$$

The uncertainties described in Eqs. 8 and 9, represent square root of the diagonal elements of $\mathbf{S_\varepsilon}(\lambda_i)$. Although it is quite common in the related information content literature to assume that there is no covariance of errors in measurements at different angles (Knobelspiesse and Nag, 2018; Chen et al., 2017; Xu and Wang, 2015), it is feasible that certain instrumentation or data processing biases influence adjacent angular measurements in a causally connected manner, thus resulting in covariant errors. To construct a more realistic covariance matrix, Knobelspiesse et al. (2012) used a *Markov process* to assign off-diagonal element values in the covariance matrix. In our analysis, we used a similar approach to account for covariance between adjacent angular measurements if the measurements are of the same type and from equal wavelengths. Furthermore, it was assumed that no error covariance exists between measurement of different types (*PF*, *PPF*) and/or different wavelengths. We employed Eqs. 10a to 10c to construct the error covariance matrix of a given measurement type at a given wavelength.

$$S_{\varepsilon,i,j} = \sigma_i^2 \qquad \text{If } i = j, \tag{10.a}$$

$$S_{\varepsilon,i,j} = \sigma_i\sigma_j\rho^{|\theta_i-\theta_j|} \qquad \text{If } i \neq j \text{ (for same measurement types and wave length)}, \tag{10.b}$$

$$S_{\varepsilon,i,j} = 0 \qquad \text{Otherwise.} \tag{10.c}$$

We set $\rho = 0.7$, for which $\rho^{|\theta_i-\theta_j|}$ varies from 0.7 at $\theta_i - \theta_j = 1°$ to below 0.05 at $\theta_i - \theta_j = 10°$, allowing for moderate covariance between adjacent angular measurements.

In the general case of a multi-wavelength configuration (with $N_\lambda$ wavelengths measurements) and for both *PF* and *PPF* measurements, the total covariance matrix will look like the block matrix demonstrated in Eq. 11, where each diagonal matrix block corresponds to the individual covariance matrices for a given type of measurement (*PF* or *PPF*) at a specific wavelength. For $\mathbf{S_\varepsilon}$, the dimension value of $N_{tot}$ is $N_\lambda \times N_\theta$ and $2 \times N_\lambda \times N_\theta$ for radiometric and polarimetric instruments, respectively.





$$S_{\varepsilon} = \begin{bmatrix} S_{\varepsilon,PF}(\lambda_1) & \cdots & 0 & & & & \\ \vdots & \ddots & \vdots & & 0 & & \\ 0 & \cdots & S_{\varepsilon,PF}(\lambda_{N_\lambda}) & & & & \\ & & & S_{\varepsilon,PPF}(\lambda_1) & \cdots & 0 & \\ & 0 & & \vdots & \ddots & \vdots & \\ & & & 0 & \cdots & S_{\varepsilon,PPF}(\lambda_{N_\lambda}) \end{bmatrix}_{N_{tot} \times N_{tot}} \qquad (11)$$

**3.3 Aerosol model, state parameters and test cases**

**3.3.1 Aerosol model and corresponding state parameters**

As an aerosol model, we generally considered aerosol particles to be homogeneous spheres with log-normally distributed size distribution modes and uniform composition within each size mode (however, a small number simulations are also performed
for non-spherical particles, as explained below). The assumption of log-normally distributed size modes is very typical in aerosol science (Seinfeld and Pandis, 2006). Furthermore, it is common to employ multi-modal log-normal functions to construct volumetric particles size distributions for a complete aerosol. This is the method adopted in this study, where Eq. 12 describes the general equation we used for a multi-modal log-normal volume particle size distribution (VPSD).

$$\frac{dV}{d\ln(r)} = \sum_{i=1}^{N_{mode}} \frac{V_i}{\sqrt{2\pi}\ln(\mathrm{GSD}_i)} exp\left[\frac{-(\ln(r)-\ln(VMR_i))^2}{2(\ln(\mathrm{GSD}_i))^2}\right], \qquad (12)$$

In this equation $N_{mode}$ is the number of modes in the size distribution. We consider unimodal ($N_{mode}$ = 1) and bi-modal size distributions ($N_{mode}$ =2). The variable $r$ is the volume equivalent aerosol radius, which for spherical particles is equal to the geometric particle radius. We assume that $r$ varies between 0.05 to 15 µm. For a given mode $i$, $VMR_i$ is the volume median
radius (equal to the geometric mean radius), $GSD_i$ is the geometric standard deviation of the distribution, and $V_i$ is the aerosol volume concentration. When using the log-normal VPSD, the size distribution related state parameters per mode $i$ are $VMR_i$, $GSD_i$, and $V_i$, meaning that the total number of size distribution related state parameters is $3 \times N_{mode}$.

We assumed that aerosol particles have uniform complex refractive index values $m$ within a given size mode. Refractive index
$m$ is a wavelength dependent property and has both a real and an imaginary component, i.e., $m(\lambda) = n(\lambda) + ik(\lambda)$, indicating that for each scattering wavelength there are two material property related state parameters, i.e., $n(\lambda)$ and $k(\lambda)$. Although we assume a uniform refractive index value for each size mode, we allow for different size modes to have different refractive index values. This leads to a total of $2 \times N_\lambda \times N_{mode}$ state parameters that are associated with refractive indices in our simulations.

Particle shape is one of the more challenging properties to incorporate in forward models of aerosol light scattering. It is commonplace to regard aerosols as spherical particles in both remote sensing and in situ light scattering applications, since



relatively simple forward models can then be constructed based on Mie theory. However, for intrinsically non-spherical particles especially in the coarse mode, such as dust particles, the spherical aerosol assumption fails to properly capture the scattering properties of aerosol particles, which results in systematic retrieval errors. Due to the lack of analytical solutions for
non-spherical particles, the aerosol light scattering has to be computed through computationally expensive numerical methods. Moreover, consideration of complex particle morphologies necessitates the use of additional state parameters to describe non-spherical particles theoretically, and at some point the total number of considered state parameters can become too high relative to the information content of the measurements.

To overcome such complexities while incorporating non-spherical particle properties, Dubovik et al. (2006) proposed the use of theoretical spheroid particles with a fixed shape distribution that is representative of the shape distribution of Saharan dust particles. Furthermore, the authors expanded the model to encompass both spherical and non-spherical particles and defined a single additional aerosol state parameter called the *spherical particle fraction* (*Sph%*). This parameter quantifies the fraction of spherical particles in an aerosol system composed of an ensemble of spherical and spheroid particles. The authors showed
that this simplified theoretical framework is suitable for modeling light scattering by a range of non-spherical particles. In this study, the majority of our simulations assume purely spherical aerosol particles. However, we also performed a limited number of test cases with the spheroid-based method of Dubovik et al. in order to assess information content in regard to the additional state parameter *Sph%*.

**3.3.2 Aerosol test cases and corresponding state parameter values ($x_0$)**

Due to the non-linear nature of the aerosol light scattering problem, the sensitivity of scattering observations varies over different values of state parameters. Therefore, for gaining proper insights through information content analysis, multiple synthetic aerosol test cases with variety of state parameters have to be assessed. These synthetic aerosol test cases should emulate generic aerosol models that are expected to be generated in real world environments.

The majority of our simulations are performed for unimodal, spherical aerosol test cases (i.e., spherical aerosols that are assumed to lie in a single size distribution mode). These aerosol test cases represent scenarios where particles are generated in a laboratory environment using, for example, size-classification instruments such as an aerodynamic aerosol classifier. (Tavakoli and Olfert, 2013). Two different aerosol materials were considered for the unimodal cases, di-ethyl-hexyl-sebacat
(DEHS) and brown carbon (BrC). DEHS is a non-absorbing oil that is used extensively to generate spherical aerosol particles and to test optical instruments. BrC is class of organic particulate matter that is moderately light absorbing at shorter visible wavelength and that occurs naturally in the atmosphere (Laskin et al., 2015). It is possible to nebulize surrogate BrC solutions in a laboratory to generate spherical BrC particles. For simplicity, we refer to the aerosol test cases with DEHS and BrC aerosols as non-absorbing and absorbing aerosol test cases, respectively. The refractive index values that were used for the



non-absorbing and absorbing aerosol test cases are presented in Table 3. For non-absorbing DEHS, we used the $n(\lambda)$ values

reported by Pettersson et al. (2004) and set $k$ to a low value of 0.0001 independent of wavelength. For the BrC we used the

values reported in Moschos et al. (2020) assuming $n$ has no spectral dependence.

**Table 3. State parameters ($x_{0,i}$) and their values for the unimodal aerosol test cases**

| | | VPSD | | | $m$ | | | |
|---|---|---|---|---|---|---|---|---|
| Material | Test cases | VMR (µm) | GSD | V (µm³/cm³) | $\lambda$ = 450 nm | $\lambda$ = 532 nm | $\lambda$ = 670 nm | $\lambda$ = 1020 nm |
| Non-absorbing aerosol (DEHS) | Fine | 0.2 | 1.4 | 8 | $1.467 + i10^{-4}$ | $1.455 + i10^{-4}$ | $1.448 + i10^{-4}$ | $1.441 + i10^{-4}$ |
| | Coarse | 1.25 | 1.4 | 8 | | | | |
| Absorbing aerosol (BrC) | Fine | 0.2 | 1.4 | 8 | $1.5 + i0.2$ | $1.5 + i0.1077$ | $1.5 + i0.0576$ | $1.5 + i0.0097$ |
| | Coarse | 1.25 | 1.4 | 8 | | | | |


Aerosol particles can take a very broad range of different sizes, and can therefore exhibit very distinct light scattering

properties. Aerosol particles with radii below 0.5 µm are often referred to as *fine*, while particles above this radius threshold

are referred to as *coarse*. To investigate the differences between differently sized particles, we defined two categories of VPSDs

labeled as the fine and coarse aerosol test cases. The VPSD-related state parameters for these two categories are shown in

Table 3. To better focus on the information content with regard to particle size, only *VMR* was varied between the fine and

coarse test cases while keeping *GSD* and *V* constant.

Ambient aerosol particles in the atmosphere typically have more complex size distributions containing multiple modes. One

common simplification in aerosol remote sensing is to use a bimodal VPSD to describe ambient aerosol size distributions,

which is typically a reasonable compromise between maintaining a sufficient level of complexity while keeping the length of

the state parameter vector reasonably small. We employed this bimodal method here in order to investigate information content

for more atmospherically relevant aerosol test cases. We defined one mode with median radius below 0.5 um as a fine mode,

and a second mode with median values larger than 0.5 µm as a coarse mode.


To construct the bimodal aerosol test cases we used size distribution and refractive index data from a study conducted by

Espinosa et al. (2019). In that study, aerosol particles where measured over wide regions of the continental USA by an aircraft-





borne laser imaging nephelometer. Aerosol state parameters (size distribution, refractive index, *Sph%*) were then retrieved

from the measurements using GRASP-OPEN. The authors reported seven distinctive classes of aerosol particles. Amongst

these classes we selected the Urban and Colorado (CO) Storm (hereafter referred to as 'Dust') cases for our study, since these

two cases represented the extreme values in terms of the fine-to-coarse mode volume concentrations and the fraction of non-

spherical particles parameter. Table 4 shows the specific state parameters that were used for the bimodal aerosol test cases.

Since in the original study only single *m* and *Sph%* values were reported for both the fine and coarse modes, we used these

values for both the fine and coarse modes in our simulations.


**Table 4 State parameters ($x_{0,i}$) and their values for the bimodal aerosol test cases. The subscripts f and c denote fine and coarse mode, respectively.**

|       | $VMR_f$ (µm) | ln($GSD_f$) | $V_f$ (µm³/cm³) | $VMR_c$ (µm) | ln($GSD_c$) | $V_c$ (µm³/cm³) | $m_f$ (all λ) | $m_c$ (all λ) | *Sph%* (Both modes) |
|-------|--------------|-------------|-----------------|--------------|-------------|-----------------|---------------|---------------|---------------------|
| Urban | 0.155        | 0.34        | 14.56           | 0.99         | 0.34        | 1.44            | 1.52 + *i*0.005 | 1.52 + *i*0.005 | 85                  |
| Dust  | 0.126        | 0.45        | 4.93            | 1.12         | 0.35        | 4.37            | 1.59 + *i*0.0043 | 1.59 + *i*0.0043 | 17                  |

The majority of the results reported in Section 4 of the present paper correspond to the unimodal aerosol test cases. Bimodal

test case results are only presented in Section 4.5, where the focus is only on insights that go beyond those obtained from the

unimodal analyses.

### 3.4 A priori covariance matrix, $S_a$

The a priori covariance matrix $S_a$ represents the expected uncertainty of retrievable state parameters before any measurement

is conducted (i.e., the assumed prior knowledge about the aerosol properties). Specification of this matrix is a critical step in

Bayesian inference, as it may or may not have a substantial effect on the inferred state (Eq. 4).  The *DOFS* metrics in Bayesian

information content analysis quantifies the relative contributions of measurement and a priori to the inferred state. Hence, it is

clear that *DOFS* also directly depends on the actual values in the a priori covariance matrix. The diagonal elements of $S_a$,

which we denote as $\sigma_{a,i}^2$, represent the a priori variances for specific state space variables. We used two different methods for

setting these:

-     Atmospheric-based method: Following the method of Alexandrov and Mishchenko, (2017), the values were based on

the ranges of previously reported ambient measurements for the given state space variables (Espinosa et al., 2019).

There ranges are listed in Table 5. There ranges were divided by two and then squared to construct variances for input

into $S_a$. Since the measurements reported by Espinosa et al. (2019) cover seven diverse classes of aerosols measured

by aircraft over the USA, we consider the ranges to be reasonably representative of continental aerosols. For our





unimodal test cases, we used only the ranges reported by Espinosa et al., (2019) for fine-mode aerosols to ensure consistency, and the same ranges were used for both non-absorbing and absorbing aerosol test cases. For the bimodal test cases we used the ranges reported for both the fine and coarse modes. This choice of a priori aims at assessing

the measurement of largely unconstrained atmospheric aerosols.

- Percentage-based method: the values were chosen to be fixed percentages of the corresponding state space parameter reference values $x_{0,i}$ (as defined in Tables 3 and 4). That is, the values were chosen such that $\sigma_{a,i}/x_{0,i} = P_i$ (%). The chosen $P_i$ values were in general different for the different types of state space parameters and they are defined in Table 5. The values were chosen based on two criteria: i) we aimed for the resulting absolute $DOFS_i$ output values to

be neither too close to 0 nor 1 across all our simulations, so that we could better identify $DOFS_i$ differences between different instrument configurations which would be obscured if retrieved state was fully dominated by measured or a priori information. ii) For some pairs of variables (e.g. the fine and coarse versions of the VPSD-related parameters such as $VMR_{fine}$ and $VMR_{coarse}$) we set equal $P_i$ values so that their corresponding $DOFS_i$ outputs could be compared in a consistent way. The motivation for the latter criterion is discussed further in the results Section 4.2. However,

simply speaking, this choice of a priori aims at providing a robust basis for comparing the information content of different polar nephelometer instrument configurations.

We set the non-diagonal elements of $\mathbf{S_a}$ to zero, which amounts to the assumption that there is no covariance between the different aerosol state parameters. This simplifying assumption is common in information content analysis (Xu and Wang,

2015; Chen et al., 2017; Knobelspiesse and Nag, 2018; Burton et al., 2016). However, it should be noted that this is indeed a simplification for certain state parameters. For example, refractive index values ($n$ or $k$) at different wavelengths are not completely independent of each other.

**Table 5. The ranges used to determine the diagonal elements of the a priori covariance matrix $S_a$ ($\sigma_{a,i}^2$). These ranges were divided by two and then squared to represent variances for input to $S_a$. The values in parentheses represent the chosen $P_i$ values for the fraction-based a priori selection method, where $\sigma_{a,i}/x_{0,i} = P_i$ (%). The two a priori selection methods are detailed in Section 3.3 of the main text.**

| Method of a priori selection | Aerosol test case | Ranges used to determine $S_a$ ($\sigma_{a,i}^2$) for parameter | | | | | | | | |
|---|---|---|---|---|---|---|---|---|---|---|
| | | $VMR_{fine}$ (μm) | ln(GSD$_{fine}$) | $V_{fine}$ (μm$^3$/cm$^3$) | $VMR_{coarse}$ (μm) | ln(GSD$_{coarse}$) | $V_{coarse}$ (μm$^3$/cm$^3$) | $n$ (all λ, fine and coarse) | $k$ (all λ, fine and coarse) | Sph % |
| Atmospheric-based | Unimodal 1 | $3.7\times10^{-2}$ | $1.3\times10^{-1}$ | 16.9 | $3.7\times10^{-2}$ | $1.3\times10^{-1}$ | 16.9 | $8\times10^{-2}$ | $3\times10^{-3}$ | - |
| | Bimodal | $3.7\times10^{-2}$ | $1.3\times10^{-1}$ | 16.9 | $4.3\times10^{-1}$ | $6\times10^{-2}$ | 3.46 | $8\times10^{-2}$ | $3\times10^{-3}$ | 51 % |
| Percentage-based | Unimodal 1 | $8\times10^{-3}$ (2%) | $2.8\times10^{-2}$ (4%)* | $3.2\times10^{-1}$ (2%) | $5\times10^{-2}$ (2%) | $2.8\times10^{-2}$ (4%) | $3.2\times10^{-1}$ (2%) | $1.4\times10^{-2}$ (0.5%) | $1\times10^{-3}$ (0.25%) | - |


*For the test cases $\sigma_{\ln(GSD)}$ of 4% is equivalent to $\sigma_{GSD}$ of ~1%

### 3.5 Forward model

To simulate the polarimetric measurement data ($PF(\theta, \lambda)$, $PPF(\theta, \lambda)$) for given aerosol test cases, the forward module of the GRASP-OPEN algorithm was employed. GRASP-OPEN is a robust and well-established inversion algorithm that has been used extensively in the aerosol remote sensing community (Dubovik et al., 2014, 2021) as well as for retrieving aerosol properties from in situ measurements (Espinosa et al., 2019). In addition to its main application as an aerosol retrieval tool, GRASP-OPEN can be run as an independent physics-based forward module. The forward module of GRASP-OPEN is quite

versatile and flexible enabling use of either binned or multi-modal log-normal representation of the size distribution in state parameter space. Another advantage of GRASP-OPEN is that it in addition to the basic, spherical particle option, it also includes the option to consider non-spherical particle shape using the spheroid-based approach and *Sph%* parameter discussed in Section 3.3.1. Therefore, all the state parameters assumed in our aerosol test cases are available in the state space implemented in GRASP-OPEN. GRASP-OPEN also provides the Jacobian matrix, **K,** of the forward model for given aerosol

states, which is required for computing information content.

### 3.6 Optimal angular sensor placement based on information content analysis

In this part of the study, we investigated to what extent particular choices of sensor placement affect the information content of polarimetric measurements. Building up on this, we introduced a method to identify optimal sensor placement for exemplary target aerosol states and instrument configurations. We constrained this analysis to unimodal, non-absorbing aerosol models

measured with single wavelength instruments with either radiometric or polarimetric capabilities. A total of $N_p = 171$ available angular positions were chosen to be at every full degree between $\theta = 5°$ to $175°$. The number of actually available sensors was then varied by considering eight cases with $N_\theta$ = 2, 3, 6, 11, 18, 35, 86, 169. This allowed us to assess the sensitivity of information content to sensor placement as a function of the number of available sensors.

Monte Carlo simulations were first conducted to randomly generate different angular sensor placements. Specifically, 1000 random draws of $N_\theta$ angles out of all the 171 available positions were made for every $N_\theta$ value. The *DOFS* values were calculated for all the randomly generated instrument configurations. This approach is not very efficient, but it does provide information on optimal sensor placement if the number of random draws is sufficiently high. As a proof-of-concept, we also developed and tested a potentially more efficient approach to obtain the optimal angular sensor placement based on the

*reductive greedy algorithm*, which is a well-known and robust optimization scheme. The greedy algorithm was applied to this optimization problem by incorporating *DOFS* calculations into the algorithm scheme. Specifically, the algorithm was set up



to selectively remove angular positions corresponding to the smallest changes in *DOFS* until only the most informative angles remained in a particular configuration. The full pseudocode for the algorithm is detailed in Supplementary Section S1.

## 4 Results and discussion

### 4.1 Dependence of information content on the angular configurations of previous polar nephelometer designs

Here we assess how the information content increases with increasing number of angular data points for the angular configurations of four polar nephelometer designs previously reported in the literature (Li et al., 2019; Nakagawa et al., 2016; Dolgos and Martins, 2014; Dick et al., 2007). For each of these angular configurations, different spectral and polarimetric capabilities are considered (specifically, all four combinations of data sets that would be obtained with single/triple wavelength

and radiometric/polarimetric measurements). The metric *nDOFS* (Table 1) was calculated for the fine, unimodal, non-absorbing aerosol test case using the atmospheric-based a priori covariance matrix given in Table 5.

Figure 3 displays the comparison of information content of data sets acquired by above instruments. As expected, the *nDOFS* (ordinate) values generally increase with increasing number of angular sensors (abscissa). However, this increase is not strictly

monotonic. Specifically, data from the four sensor instrument has comparable or even slightly higher *nDOFS* compared to the seven sensor instrument depending on which spectral/polarimetric capabilities are considered. This suggests that implementing more sensors on its own is not necessarily enough to achieve more informative data and that the particular placement of the sensors may also be important. Comparable performance of the four and seven sensor instruments for the test aerosol case assessed here should not be interpreted in terms of better or poorer instrument design. Instead, the four sensor instrument may

have been optimized for probing aerosols similar to the test aerosol, whereas the seven sensor instrument may have been optimized for different target aerosol properties. The significance of sensor placement for different test aerosol properties and variable sensor number is explored in more detail in Section 4.8.

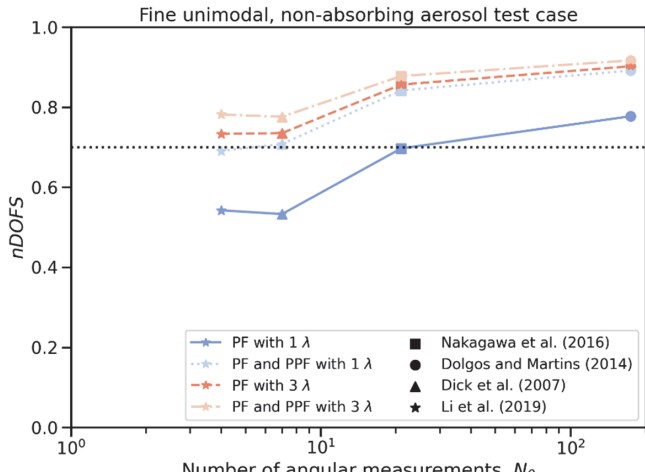

**Figure 3. Variation of *nDOFS* for spectral polar, polarimetric, and angular configurations. The angular configurations are based on instruments available in the literature. The dotted line corresoonds to *nDOFS* equals 0.7 and is included for reference.**

Figure 3 also shows that the addition of spectral and/or *PPF* measurement capabilities can result in *DOFS* increases that are comparable to the increases obtained by simply adding more angular sensors. For example, it can be seen that for this aerosol test case, addition of  polarization and/or multi-wavelength measurements will make the *nDOFS* values of instruments with four and seven angular sensors roughly equivalent to those of a single wavelength instrument with 21 sensors without polarimetric measurement capabilities. In the following sections the effect of polarimetric, spectral and angular measurement on information content of a variety of retrievable state parameters are explored in more detail.

**4.2 Dependence of information content on the available prior knowledge of the aerosol state parameters**

Polarimetry may be applied to a wide range of different aerosol types and in applications with different prior knowledge on the aerosol under investigation. Qualitatively speaking, the more prior knowledge is available on the state parameters of a certain aerosol, the more challenging it becomes to gain additional information through probing this aerosol with a polarimetric measurement. The *DOFS*-based information content metrics as introduced in Section 2 are one approach to quantify differences in information gain as a function of prior knowledge, instrument configuration, measurement error and reference state of the probed aerosol (Fig. 1). Here we focus on the effect of prior knowledge on *DOFS*, which comes in through the a priori variance matrix (Eqs. 3 and 4), and how to interpret and compare resulting *DOFS* values. For this purpose we choose two distinct variants of a priori knowledge as detailed in Section 3.4. As for the measurement capabilities, we computed $DOFS_i$



over different $N_\lambda$ values ($1\lambda$, $3\lambda$, $4\lambda$; see Section 3.1.4) and polarimetric settings (radiometric or polarimetric) for a fixed angular configuration. Specifically, we selected a generic angular configuration with $\alpha = 5°$, $\beta = 0°$, and $\Delta\theta$ of $10°$ (which corresponds to $N_\theta = 18$) to represent a polar nephelometer with moderate angular sensor resolution. We used the fine and coarse unimodal, non-absorbing aerosol test cases for this investigation (see Section 3.3.2).

To address the measurement of largely unconstrained atmospheric aerosols, $DOFS_i$ values were calculated using the atmospheric-based a priori covariance matrix detailed in Table 5. That is, the aerosol state parameters were a priori assumed to vary over the ranges covered by the comprehensive ambient measurements reported by Espinosa et al. (2019). Figure 4 displays the $DOFS_i$ results as a function of instrument capabilities as well as for fine and coarse test states of a unimodal aerosol. The results indicate that all the tested configurations provide high information content in this situation. For example,

even the $1\lambda$ instrument produces $DOFS_i$ values above 0.8 for the majority of the unimodal aerosol state parameters (with the exceptions of $VMR_{coarse}$, $\ln(GSD)_{fine}$, and $k_{fine}$). Values of $DOFS_i$ greater than 0.8 indicate that the posterior value of a given retrievable parameter is driven almost entirely by the information a new measurement provides, rather than what is known a priori about that value. In this sense, these results demonstrate that even a single wavelength polar nephelometer with moderate angular resolution will provide very informative measurements in relation to the expected variability in ambient aerosol

properties. However, while this is a useful insight, the overall high $DOFS_i$ values shown in Fig. 4 (i.e., very close to one) impede the comparison of the different spectral and polarimetric instrument configurations.



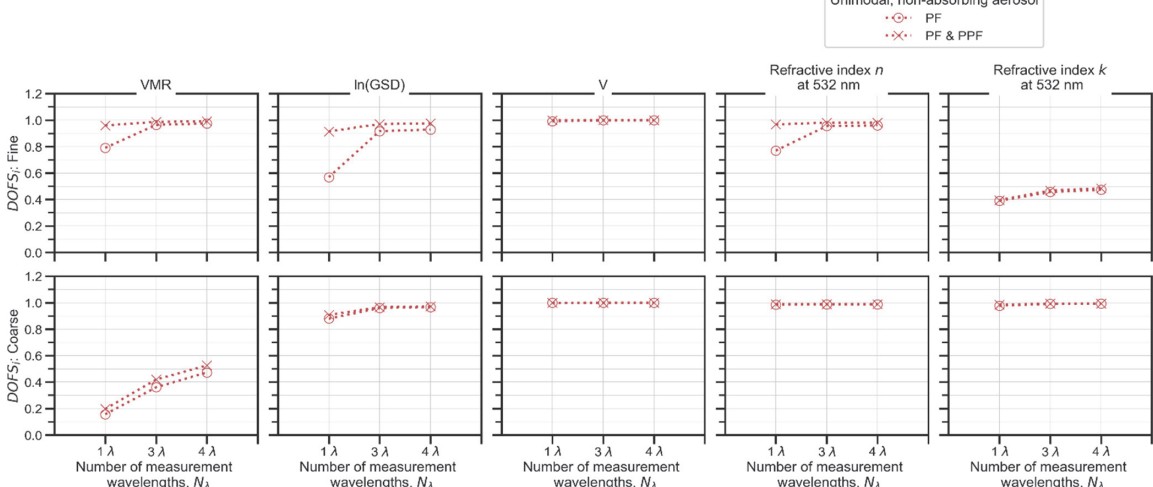

**Figure 4: $DOFS_i$ values for specific aerosol parameters corresponding to the fine (upper row) and coarse (bottom row) unimodal,**
**non-absorbing aerosol test cases. The a priori variance values are based on the atmospheric method described in Section 3.3 and the**
**measurements from Espinosa et al. (2019).**

The main reason that the $DOFS_i$ are quite high over most of the state parameters is due to the assumption for the a priori

variance values. $DOFS_v$ is a prime example of a state parameters with $DOFS_i$ very close to 1 for both fine and coarse cases.

Such results suggest that even a basic polar nephelometer can provided ample information on aerosol properties when measured

against prior knowledge that is as crude as: "unimodal atmospheric aerosol with model diameter somewhere in fine or coarse

mode". This is not surprising given that the atmospheric-based a priori variance for e.g. the state parameter $V$ ($\sigma_{a,V}$ = 8.5

µm³/cc; Table 5) is larger than the actual reference value of $V$ (8 µm³/cc; Table 3) considered for all the unimodal test cases.

It is worth mentioning that the $\sigma_{pro}^2/\sigma_a^2$ results which are depicted in Fig. S1, show that for parameter $V$ the $\sigma_{pro}^2/\sigma_a^2$ decreases

by a factor of ~ 10 when the number of measurement wavelengths is increased from 1λ to 3λ. This demonstrates that despite

the very high $DOFS_v$ values already achieved by a single wavelength instrument, addition of spectral capabilities does actually

substantially reduce the uncertainty of the retrieved state.

Results from Fig. 4 further suggest that the $DOFS_{VMR}$ values for the fine case are systematically larger than those for the coarse

case. In the $DOFS_i$ calculations of Fig. 4, the absolute a priori variances for $VMR$ in fine and coarse cases are identical. This

means that relative to the given fine and coarse $VMR$ state values (0.2 and 1.25 µm, respectively), the a priori variance fraction

is much greater for the fine than the coarse case. Therefore, the finding that $DOFS_{VMR}$ is higher when probing fine mode

compared to probing coarse mode within a given a priori range must not be interpreted as better performance relative to the


retrieved *VMR* state values. An alternative approach, to make $DOFS_{VMR}$ more comparable for different tested aerosol states, is to assume that the a priori variance is percentage of the corresponding state parameter value (percentage separately fixed for each state parameter at values listed in Table 5). Figure S2 presents $DOFS_{VMR}$ results for the two alternative a priori variances: that is, either the atmospheric range, or as fixed percentage of tested state. Indeed, it can be seen that by using percentage-based a priori variance, the $DOFS_{VMR}$ values and trends with spectral capabilities become similar for fine and coarse test cases.

These results imply that the polar nephelometer measurement is comparably informative in relative terms for retrieving VMR for both fine and coarse mode aerosol.

         Using low percentages of the test case state parameter values as corresponding a priori ranges sets tighter bonds on the a priori knowledge compared to the wider atmospheric a priori range. This results in lower information content of identical

measurements when expressed with $DOFS_i$. A side effect of shifting $DOFS_i$ down to medium values is that $DOFS_i$ becomes more sensitive to differences in instrument configurations and measurement precision for equal aerosol state test cases. Therefore, we chose the percentage-based a priori ranges for computing $DOFS_i$ in most of the following examples. Furthermore, the percentages, as listed in Table 5, were chosen at levels such that the majority of $DOFS_i$ values over all instrument configurations and tested aerosol states does not assume extreme values near 0 nor near 1.


         Figure 5 is equivalent to Fig. 4 but for showing $DOFS_i$ values for percentage-based a priori ranges for the unimodal aerosol model. One of the very basic insights that can be drawn from this figure is that for given state parameters, $DOFS_i$ always increases when the measurement configuration becomes more complex, be it through additional measurement of *PPF* or measurements at multiple wavelengths. However for some related pairs of data points, the $DOFS_i$ increases can be very minor.

For example in the coarse mode case, $DOFS_n$ increases by only 0.0005 when the number of measurements wavelengths is increased from 3λ to 4λ for a radiometric measurement (i.e., *PF* only). This indicates potentially poor cost to benefit ratio of performing more complex measurements for accurate quantification of the corresponding state parameter.

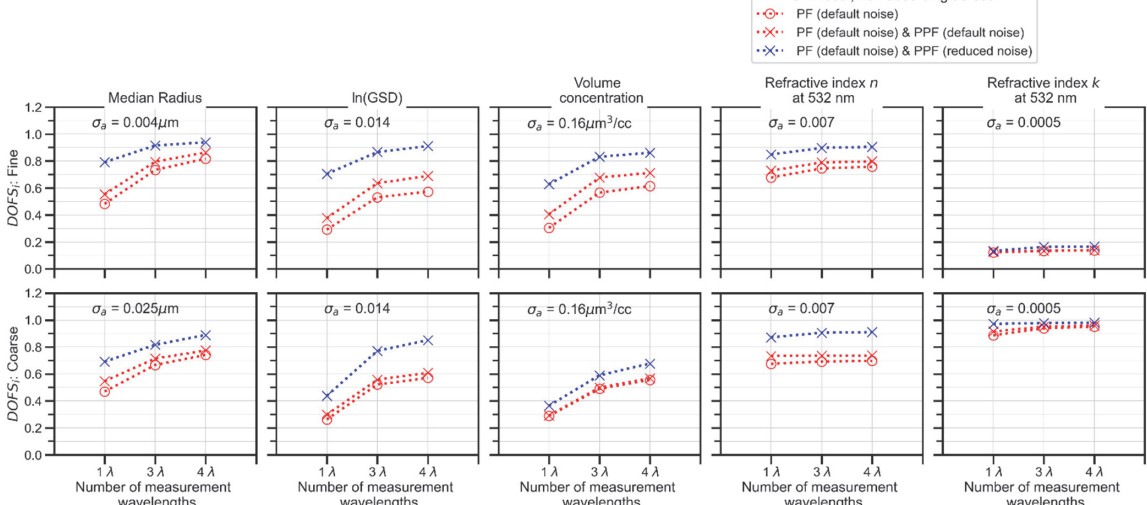

**Figure 5.** *DOFS_i* **values for specific aerosol parameters corresponding to the fine (upper row) and coarse (bottom row) unimodal, non-absorbing aerosol test cases. The a priori variances ($\sigma_a$) for each state space parameter are displayed as text boxes in each subpanel. These values were chosen according to the percentage-based method described in Section 3.3. That is, the a priori variances correspond to fixed percentages of the respective state space values ($x_{0,i}$). The blue lines are the *DOFS_i* values corresponding to a reduced *PPF* measurement noise level of 0.01, while the default noise is 0.056 (Eq. 9). The relative *PF* measurement noise level is 0.046 (Eq. 8) for all curves.**

Figure 5 also indicates that when the a priori variances are chosen as fixed percentage of the corresponding $x_{0,i}$ values, then the trends and values of $DOFS_i$ between fine and coarse test cases are quite similar for all aerosol parameters except the imaginary part of the refractive index $k$. This suggests that the finding from Fig. S2 for the *VMR* state parameter applies more generally to most state parameters: the polar nephelometer measurement is comparably informative in relative terms for most retrievable aerosol state parameters independently of whether one is probing fine or coarse unimodal aerosols. The exceptional behavior of $DOFS_k$ in this regard is explained in more detail in Section 4.4.

### 4.3 Information content gain associated with more comprehensive measurements and reduced measurement error

In Section 4.2 we explained that $DOFS_i$ calculated with percentage-based a priori knowledge, representing a polarimetric problem with ample prior knowledge on the aerosol state parameters, is a sensitive metric to demonstrate benefits of more comprehensive or more precise polarimetric measurements. Therefore, the $DOFS_i$ results based on this variant of a priori knowledge that are presented in Fig. 5 are used for the following discussion.



### 4.3.1 Information content gain due to multi-wavelength measurements

Figure 5 indicates that increasing the number of measurement wavelengths can lead to noticeable information gain for all three size distribution parameters (mean radius, GSD, and volume concentration). For each of these parameters, $DOFS_i$ for the 3λ instrument configurations are considerably higher compared to the single wavelength configurations. For example, $DOFS_{ln(GSD)}$ and $DOFS_V$ increase by an absolute value of 0.3 (30% of full $DOFS_i$ scale) when going from a single to a 3λ instrument (for both fine and coarse cases).

The results further indicate that although the addition of infrared wavelength in the 4λ also increases the $DOFS_i$ values, such improvement is noticeably smaller than the $DOFS_i$ improvement when transitioning from 1λ to 3 λ configuration. For most of the test cases, the increase in information content when moving to a 4λ configuration was less than ~ 50% of the corresponding $DOFS_i$ increase when transitioning from 1λ to 3λ. This observation suggests that the addition of a fourth wavelength measurement in the near IR is not very beneficial for the retrieval of the size distribution parameters of a unimodal, homogeneous and spherical aerosol.

For the refractive index parameters, Fig. 5 suggests that the addition of multi-wavelength measurements only weakly increases information content with respect to the *n* and *k* values at the wavelength of the single wavelength instrument (532 nm in this case). However, it should be noted that multi-wavelength measurements do increase information content in the sense that they enable the retrieval of refractive index values at the additional wavelengths that the measurements are performed at (e.g. this is shown for $DOFS_n$ values at 450, 532, 630, and 1020 nm in Fig. S3). This is a trivial result that simply expresses the expectation that adding measurement wavelengths to a given instrument configuration brings information about the refractive index values at the added wavelengths.

### 4.3.2 Information content gain due to polarimetric measurements and reduced measurement error

Comparing the curves with red circles and red crosses in Fig. 5 provides insight into the information content benefit of adding *PPF* measurements with an absolute error of 0.056 (Eq. 9) This level of absolute *PPF* measurement error is based on the laser imaging nephelometer of Dolgos and Martins (2014). For this case, addition of *PPF* results in only minor $DOFS_i$ increase (at most ~ 0.1) across all the λ configurations for fine mode volume concentration and ln(*GSD*). For all other parameters the $DOFS_i$ increases were even lower. By contrast, the corresponding $DOFS_i$ increases when transitioning from a 1λ to a 3λ instrument configuration are considerably larger.

In the context of remote sensing instrumentation, *PPF* measurement errors can be considerably lower than the default noise level of 0.056 that is considered here. For example, Xu and Wang (2015) consider an absolute error of 0.01 for polarized measurements made by AERONET sun photometers. The blue curves in Fig. 5 demonstrate the information content benefit of





adding *PPF* measurements with such reduced absolute error (e.g. 0.01 rather than 0.056) while retaining the *PF* noise level unchanged. This change results in substantial information content increase over all state parameters and wavelength numbers, with the exception of *V* (coarse) and *k* (coarse and fine). With the reduced level of noise in the *PPF* data, a single $\lambda$ instrument

with polarimetric capability could be similarly or even more informative than a $4\lambda$ instrument with *PF* measurement only (e.g. see the $DOFS_i$ values for the size distribution parameters of the fine test case in Fig. 5).

It should be noted that all of the results shown so far are valid for probing rather simple aerosol systems: i.e., unimodal size distributions of spherical particles with homogeneous optical properties. For more complex aerosols the information content

benefit associated with the addition of polarimetric measurements can be even higher, as we demonstrate below in Section 4.5.

**4.4 Information content for the imaginary part of the refractive index**

Light absorbing aerosols are found ubiquitously throughout the atmosphere. Retrieving information on light absorption by an aerosol from polarimetric light scattering data is known to be difficult and can require the use of additional independent measurements as constraints (e.g. Espinosa et al., 2019). Nevertheless, the results shown in Fig. 5 for $DOFS_k$ demonstrate that

some information can be retrieved. In the following we assess how this depends on actual aerosol test case.

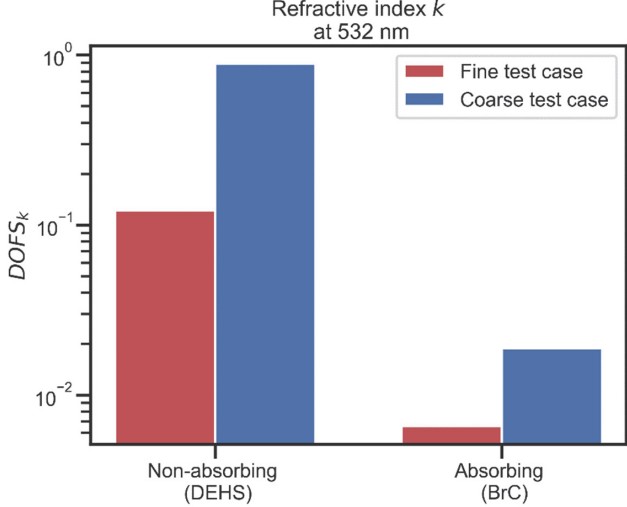

**Figure 6.** $DOFS_k$ at 532 nm for non-absorbing and absorbing, and fine and coarse mode aerosol test cases. In all cases spherical particles are assumed and a $1\lambda$ *PF* only measurement configuration.






First we assess the difference in $DOFS_k$ between non-absorbing and mildly absorbing BrC test case aerosols (Table 3) for the unimodal aerosol model. Fig. 6 plots these different $DOFS_k$ values at 532 nm for a single-wavelength, $PF$ only measurement configuration. For this configuration, the $DOFS_k$ values for non-absorbing aerosols (0.12 for fine mode, 0.89 for coarse) are

significantly larger than the corresponding values for absorbing aerosols (0.006 for fine mode, 0.019 for coarse mode). Such noticeable systematic difference implies that the scattering measurement is more informative regarding the exact value of $k$ for non-absorbing particles compared to absorbing particles. Secondly, the results suggest that there are considerable differences in $DOFS_k$ between the coarse and fine aerosol cases, i.e. the polarimetric light scattering measurement is much more informative regarding $k$ for coarse aerosol compared to fine aerosol (using otherwise equal aerosol state parameters).

Although the results in Fig. 6 only pertain to a 1$\lambda$ instrument configuration, similar results are also obtained for multi-wavelength instrument configurations as shown in Fig. S4a.

To explore this finding in more detail, we assess the sensitivity of phase function to $k$ using the absolute value of the error-corrected Jacobian ($K_{EN, i}$), which was introduced by Xu and Wang (2015) for a given state parameter $x_i$ based on the following

formula:

$$K_{EN,i} = |\left(\frac{\partial y}{\partial x_i}\right) \times \left(\frac{\sigma_{a,i}}{\sigma_{meas}}\right)|, \tag{13}$$

where $\mathbf{y}$ is either the $PF$ or $PPF$ measurement vector, $\sigma_{a,i}$ is the square root of a priori variance for parameter $x_i$, and $\boldsymbol{\sigma}_{meas}$ is the vector of measurement uncertainties. $K_{EN,i}$ can be interpreted as a normalized partial derivative of $PF$ (or $PPF$) with respect to $x_i$, which is a measure of the measurement sensitivity to perturbation in state parameter $x_i$. Figure S4b shows the results of

$K_{EN,k}$ as a function of scattering angle for the non-absorbing and absorbing aerosols. The results indicate that for a given size test case (coarse or fine), the normalized derivatives of the non-absorbing aerosol test cases are systematically larger than those for the absorbing cases. The results also indicate that for a given aerosol type (non-absorbing or absorbing), the coarse aerosol cases have larger normalized derivative values than their fine counterparts at the majority of scattering angles.

Although $K_{EN,k}$ as displayed in Fig. S4 is useful for gaining an overview on measurement sensitivity to state parameter perturbation, the results are not always sufficient for interpretation of $DOFS$ and information content. This is due to the fact that the Bayesian approach accounts for inter-dependence of information content across different state parameters. However, the $DOFS_i$ of parameters other than $k$ is rather insensitive to changing $k$ in the aerosol test case. This can be seen when comparing Figs 5 and S5, where latter is equivalent but for assuming a $k$ value representative of absorbing BrC aerosol instead

of zero. Therefore, using $K_{EN,k}$ should lead to comparable results as the $DOFS_k$ metrics. Indeed, both Fig. S4 and Fig. S5 are consistent with the general findings that $DOFS_k$ decreases with decreasing particle size and with increasing $k$.





### 4.5 Information content for bimodal non-spherical aerosol model

The more complex an aerosol, the more state parameters are required to describe it. Retrieving the state parameters of more
complex aerosols from polarimetric measurements will, accordingly, require data with overall higher information content. This
may amplify the benefit of performing measurements with instruments that have more comprehensive capabilities or smaller
measurement errors. To address this hypothesis, we used the bimodal non-spherical aerosol model with two test cases based
on the results from Espinosa et al. (2019). These two cases represent 'urban' and 'dust-dominated' aerosols measured over the
USA. The *DOFS* values are calculated using the environmental-based a priori variance assumptions and the default noise level
for *PPF* (i.e., 0.056). The results are shown in Fig. 7. It should be reiterated that for these analyses the a priori variances
assumed for coarse and fine size distribution parameters are not identical. For example, the a priori variance for coarse median
radius is ~ 130 times larger than the fine median radius. Hence, $DOFS_i$ should not be compared in absolute terms between fine
and coarse test cases.

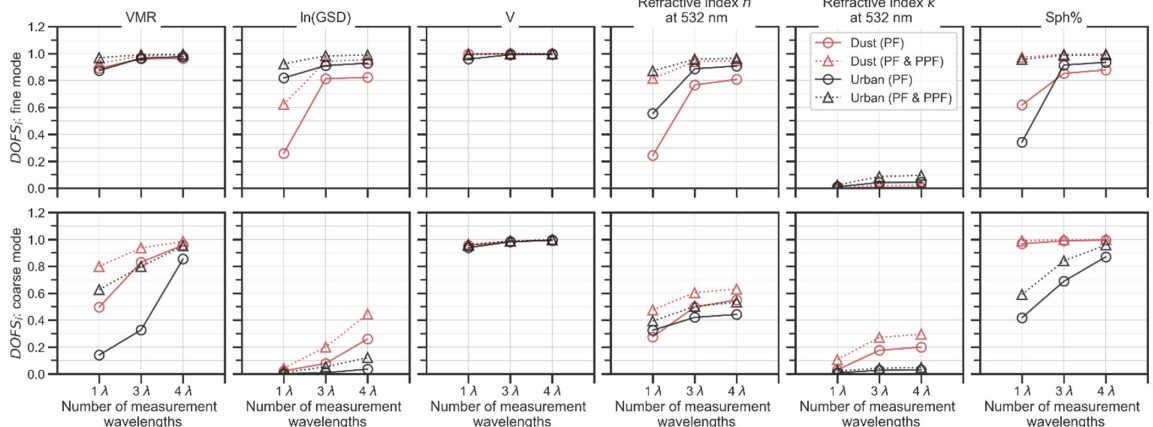


**Figure 7: $DOFS_i$ values for specific aerosol parameters corresponding to the fine (upper row) and coarse (bottom row) bimodal non-spherical aerosol model with a priori variance values based on results from Espinosa et al. (2019)**

The $DOFS_i$ results from the bimodal test cases indicate that both multi-wavelength and *PPF* measurements can greatly increase
the retrievable information about aerosol shape. For example, the addition of *PPF* measurement (triangle markers) results in
$DOFS_{Sph\%}$ for fine modes values becoming greater than 0.9 for both aerosol test cases, even though the *PPF* measurement
error is assumed to be high. The effect of *PPF* measurement on $DOFS_{Sph\%}$ is weaker in coarse aerosol. On the other hand, the
transition from 1λ to 3λ can also increase the $DOFS_{Sph\%}$ considerably, for example up to differences of 0.3 for coarse mode


and 0.6 for fine mode. The 4th $\lambda$ further increases the $DOFS_{Sph\%}$ unless values near one are already reached with the three $\lambda$ variant.

The results further show strong $DOFS_n$ improvements (fine and coarse) when transitioning from $1\lambda$ to $3\lambda$, while the benefit starts to level off at the 4th wavelength. Furthermore, the benefit of adding $PPF$ measurements is also highlighted by the substantial $DOFS_i$ increases achieved for $n$ as well as the size distribution parameters ($VMR$, ln($GSD$)), especially for the $1\lambda$ instrument configuration. Overall, the complexity of the bimodal aerosol model including a non-spherical shape parameter ($Sph\%$) raises the polarimetric challenge to a level where performing more comprehensive measurements in terms of spectral and/or polarimetric capabilities clearly benefits the achieved information content, and hence the retrievability of state parameters.

**4.6 Dependence of information content on angular measurement truncation**

As described in Section 3.1, angular truncation in nephelometry refers to the inability to measure light scattering at certain angles due to physical design limitations. To investigate the effect of truncation on aerosol property retrieval potential, we used two simplified aerosol tests cases (fine and coarse unimodal, non-absorbing, spherical particles) and two measurement configurations: a basic instrument ($\Delta\theta=1°$, $1\lambda$ and $PF$ only), and a comprehensive instrument ($\Delta\theta=1°$, $4\lambda$ and $PF$ & $PPF$). For all of these cases, we employed the default measurement noise setting (e.g., relative $PF$ measurement error of 0.046, and absolute $PPF$ measurement error of 0.056).

**4.6.1 Extreme angle truncation**

Figure 8 demonstrates the effect of extreme angle truncation $\alpha$ (i.e., loss of light scattering data at extreme forward and backward angles) on $DOFS_i$ for the five unimodal aerosol state parameters. As expected, in all of the simulated cases, $DOFS_i$ decreases as $\alpha$ increases and measurement information is lost. Generally, the $DOFS_i$ values for the fine test cases decrease less than the corresponding values for the coarse test cases. For the fine case with basic instrument configuration, only $DOFS_{VMR}$ and $DOFS_k$ display notable sensitivity to $\alpha$, decreasing by ~0.4 and ~0.15, respectively, when $\alpha$ is increased from 0° to 40°. The extreme angle truncation effect on $DOFS_{VMR}$ can even be largely avoided by use of the comprehensive $4\lambda$, polarimetric instrument configuration. However, this more comprehensive instrument configuration has little effect on the truncation effect for $DOFS_k$. Overall, the relative insensitivity of fine-mode $DOFS_i$ (excepting $DOFS_k$) to the loss of measurement information at forward and backward angles can be understood by the fact that the scattering phase functions of fine-mode aerosols are relatively isotropic, with no distinct features in the extreme angle scattering directions.





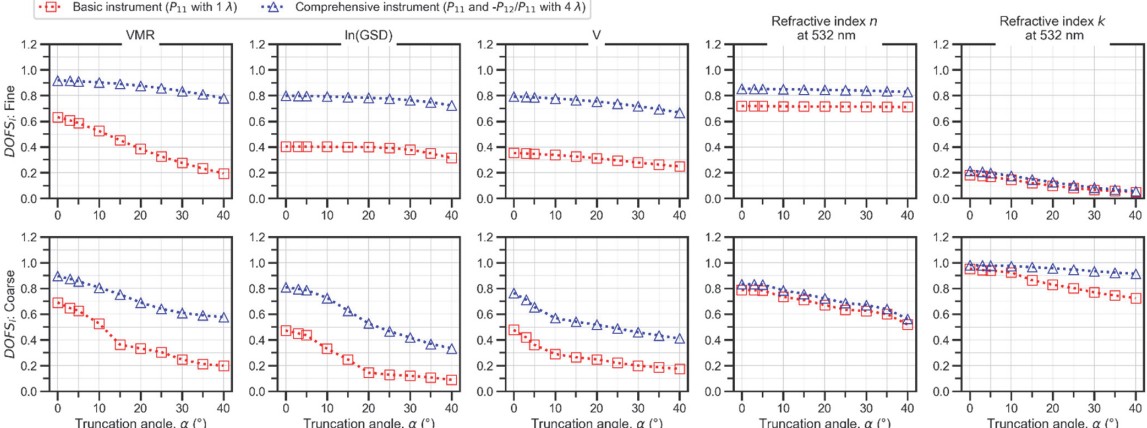

**Figure 8: Variation of *DOFS*i for unimodal, non-absorbing aerosol test cases over different extreme angle truncation cases. The top row demonstrate the results for fine aerosol test case while the bottom row corresponds to results for coarse aerosol test case.**

In contrast to the fine test case, Fig. 8 indicates that $DOFS_i$ for the coarse test case is relatively sensitive to $\alpha$ for all five unimodal aerosol state parameters. For the basic instrument configuration, $DOFS_i$ for all state parameters decreased by at least 0.15 when $\alpha$ was increased from 0˚ to 40˚. Furthermore, the addition of multi-wavelength and polarimetric capability with the comprehensive instrument configuration did little to change these trends, excepting for the state parameter $k$ where a noticeable improvement was observed.  The differing response to extreme angle truncation for simple coarse-mode aerosols compared to

fine-mode aerosols can be explained by the fact the phase functions of spherical, coarse particles are relatively more forward-focused than those of spherical, fine particles, and therefore, the loss of measurement information at extreme forward angles has a greater effect on the retrievability of the coarse aerosol properties.

       This information content analysis of the extreme angle truncation effect has implications for polar nephelometer design. For

example, if an instrument is being designed to measure both fine and coarse aerosols, then minimizing extreme angle truncation as much as technically possible is a beneficial aim. However, if an instrument is being designed to measure only simple, fine aerosol particles (i.e., without complex shapes), than one should aim to include multi-wavelength and polarimetric measurement capabilities rather than expending considerable design effort on minimizing extreme angle truncation.

**4.6.2 Side scattering truncation**

As explained in Section 3.1.2, some polar nephelometer designs can also suffer from side angle truncation (i.e., the loss of scattering information at scattering angles around 90˚, which we denote with the angle $\beta$). Fig. 9 demonstrates the effect of side angle truncation on $DOFS_i$ for the five unimodal aerosol state parameters. For all five of the parameters, only relatively



minor decreases in $DOFS_i$ (< 0.2) are observed when $\beta$ is increased, even when it is set to very large values of 40°. This finding is valid for both fine- and coarse-mode aerosols, and for both basic and comprehensive (i.e., polarimetric) instrument
configurations. This suggests that in contrast to extreme angle truncation (Fig. 8), side angle truncation has little effect on the ability to retrieve fine- and coarse-mode aerosol properties from polar nephelometer data.

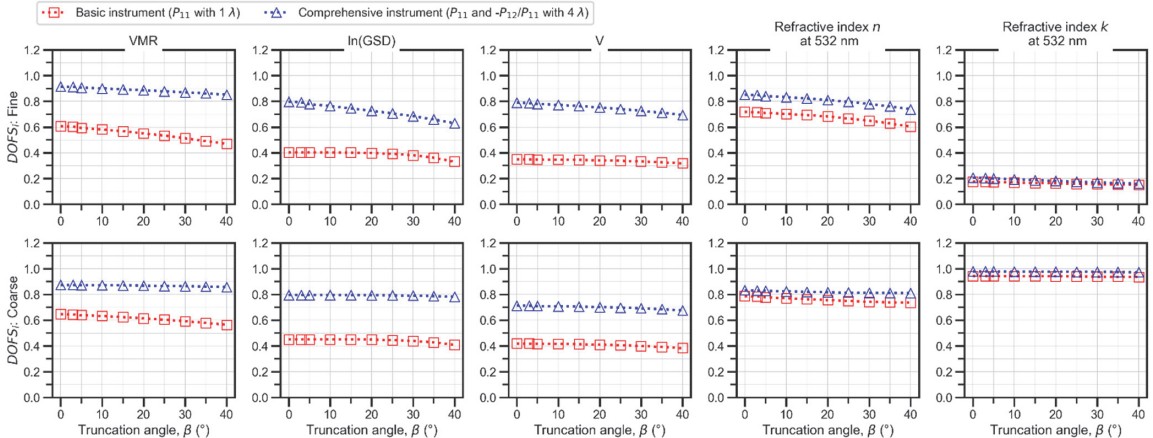

**Figure 9: Variation of $DOFS_i$ for unimodal non-absorbing aerosol test cases over different side-truncation angles. The top row**
**demonstrate the results for fine aerosol test case while the bottom row corresponds to results for coarse aerosol test case**

**4.7 Dependence of information content on the number of angular measurements**

The number of angular measurements, $N_\theta$, is a central design feature of polar nephelometer instruments. $N_\theta$ can vary greatly – from 4 or lower for fixed detector systems (Nakagawa et al., 2016), up to 174 for laser-imaging nephelometers (Dolgos and
Martins, 2014) – and it has substantial impact on the overall information content of polarimetric data as discussed in Section 4.1 using the metric $nDOFS$. Here we assess the effect of different $N_\theta$ on information content regarding individual aerosol state parameters in more detail using the $DOFS_i$ metrics. For this purpose, we consider eight different angular sensor configurations with $N_\theta$ ranging from 2 to 171. In each of the eight configurations, the angular sensor locations are evenly distributed (see Section 3.1.3 and Table 2). As in the previous Section 4.6, for these simulations we used two simplified aerosol tests cases
(fine- and coarse-mode unimodal, non-absorbing, spherical particles), two measurement configurations (a basic instrument with 1$\lambda$ and $PF$ only; and a comprehensive instrument with 4$\lambda$ and $PF$ & $PPF$; and in each case $\alpha = 5°$ and $\beta = 0°$), default measurement noise values, and the percentage-based a priori variances.





Figure 10 displays $DOFS_i$ as a function of the number of angular measurements $N_\theta$ for five unimodal aerosol state parameters

(3 size distribution parameters and 2 refractive index parameters for the wavelength of the 1λ instrument). For all of the
parameters, $DOFS_i$ increases as $N_\theta$ increases, i.e. all $DOFS_i$ contribute to the increase of overall $nDOFS$ with increasing $N_\theta$ as
shown in Fig. 3. Interestingly, the $DOFS_i$ values tend to begin plateauing out at $N_\theta$ values of around 20 – 40. This indicates
that for the investigated combination of relatively simple aerosol models, high a priori knowledge (i.e., small a priori variance),
and default measurement noise, excessive angular resolution does not result in substantial information gain. If $N_\theta$ already falls

into the plateau range, then information content can still be increased by addition of multi-wavelength and polarimetric
measurement capabilities (blue curves in Fig. 10). Based on results in the previous sections, it can be expected that the plateau
in information content as a function of $N_\theta$ only kicks in at higher $N_\theta$ for e.g. more complex aerosol models or smaller
measurement noise. However, quantitative validation of this hypothesis is beyond the scope of the current work.

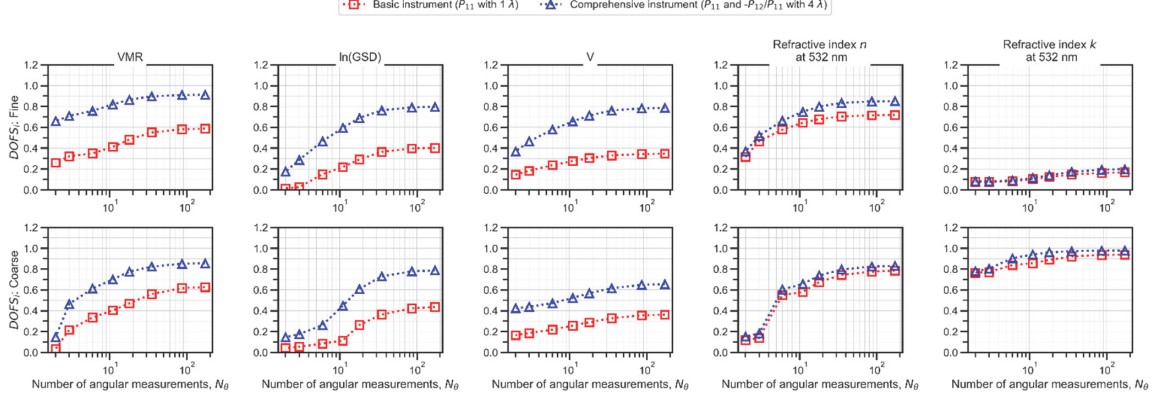


**Figure 10: Variation of $DOFS_i$ for unimodal, non-absorbing aerosol test cases as a function of the number of angular measurements $N_\theta$. The top row demonstrate the results for fine aerosol test case while the bottom row corresponds to results for coarse aerosol test case. These results were obtained using default measurement noise values, and the percentage-based a priori variances.**

Finally, it should be noted that for some of the instrument configurations considered in this $N_\theta$ analysis, the dimension of
observation become smaller than the number of retrievable parameters, resulting in an underdetermined inverse problem. This
is noticeable for $N_\theta < 6$, where drastic reductions in $DOFS_i$ values are observed for all state parameters, especially for the basic
instrument configuration with $PF$ only at 1λ. For such underdetermined cases, the maximum possible value of the sum of
$DOFS_i$ over all state parameters is equal to the observation dimension rather than the number of retrievable parameters

(Rodgers, 2000). For example, in Fig. 10, it can be seen that for the basic instrument configuration with $N_\theta = 2$, the sum of all
fine-mode $DOFS_i$ values is 0.81 ($DOFS_{VMR} = 0.26$, $DOFS_{ln(GSD)} = 0.014$, $DOFS_V = 0.145$, $DOFS_n = 0.316$, $DOFS_k = 0.075$),
which is less than the observation dimension of 2.





**4.8 Proof of concept for using *DOFS* as metric for optimizing angular sensor placement**

In the previous Section 4.7, $N_\theta$ was varied while assuming that the angular measurements were evenly distributed across the full range of scattering angles $\theta$. Equidistant sensor placement has often been applied to the design of fixed-detector-type polar nephelometers (e.g. Nakagawa et al., 2016). However, one might expect that for low values of $N_\theta$, the specific locations of the different angular measurements/sensors within the scattering angle range will also be an important handle to maximise information content. To explore the potential benefit of optimized angular sensor placement, we conducted a Monte Carlo

simulation where we computed *nDOFS* for 1000 different random sensor placement configurations given eight different $N_\theta$ values (see Section 3.5). For this analysis we used a basic aerosol model (unimodal size distribution) and assessed two state cases (fine and coarse, non-absorbing, spherical particles), and one measurement configuration corresponding to the basic polar nephelometer design (single wavelength, *PF* and *PPF*).


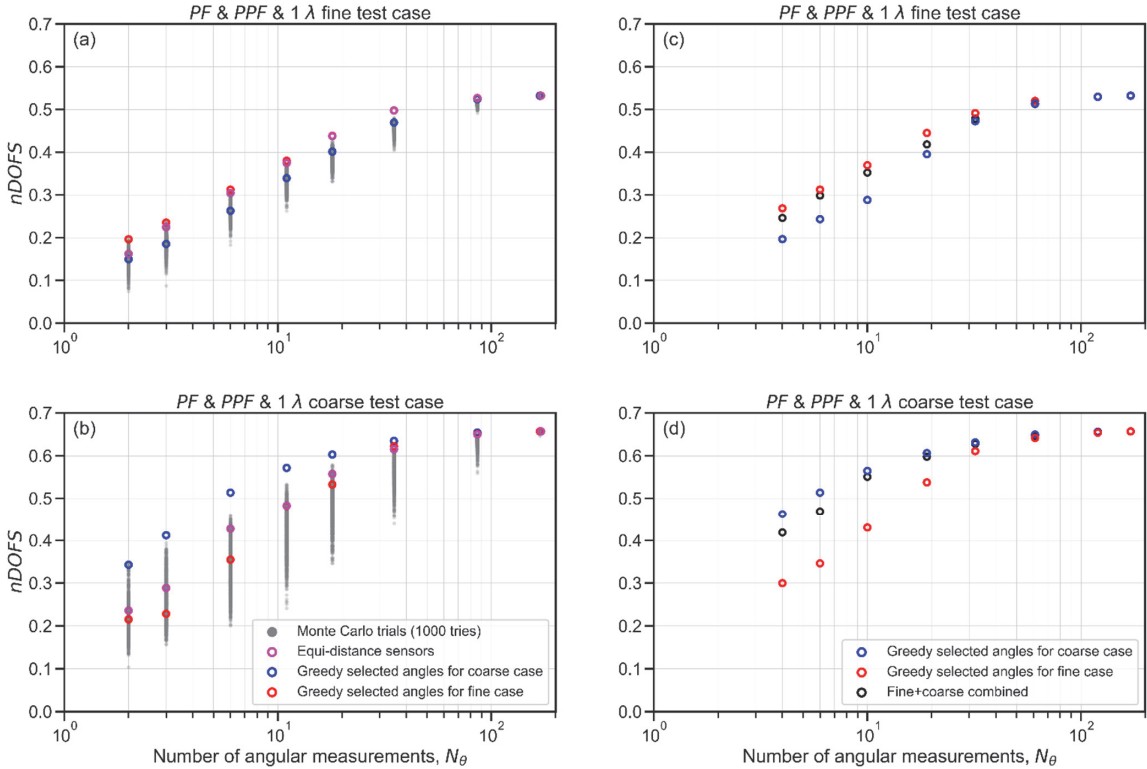





**Figure 11. (a) and (b) shows the *nDOFS* values for optimized angular configurations for different sensor numbers based on reductive greedy algorithms. The red points correspond to optimized configurations of fine aerosol test case while the blue points corresponds to optimal angular sensor configuration of coarse aerosol case. The grey dots are results of Monte Carlo simulations of 1000 random angular configurations for each case of angular sensor numbers. In (c) and (d) the black dots correspond to the normalized *nDOFS* values for the combined optimal configurations of fine and coarse test cases**

Figures 11a and b show the results of the Monte Carlo simulations as grey markers for both the fine and coarse test cases, respectively. At a given $N_\theta$, the grey markers spread over a range of *nDOFS* values. For example, for the coarse test case with $N_\theta = 11$, *nDOFS* varies from $\sim 0.2$ to $0.5$ across the 1000 randomly generated sensor placements, which implies that choosing the optimal sensor placement could substantially increase information content and thus retrievability of state parameters. Generally, this spread in *nDOFS* values becomes greater as $N_\theta$ decreases. This confirms the suppositions that, first, actual sensor location is important and, second, the potential benefit of optimizing sensor location increases with decreasing values of $N_\theta$. Furthermore, the Monte Carlo results demonstrate that the spreads in *nDOFS* values are smaller for the fine-mode test cases than the corresponding coarse test cases. Therefore for a given $N_\theta$ value, choosing the right sensor placement is more critical if one seeks to measure coarse rather than fine particles.

While the Monte Carlo method suggests a brute force approach to the task of identifying optimal sensor placements, we also present here a proof of concept analysis showing how a conventional optimization method could be applied to this problem. For this purpose we employ the reductive greedy algorithm described in Section 3.5. The *nDOFS* values corresponding to the sensor placements determined by the greedy algorithm are also displayed in Fig. 11. The algorithm was applied separately to the fine-mode (red open markers) and coarse-mode (blue open markers) test cases. To place these results in further context, the corresponding *nDOFS* values for equidistantly placed sensors are also displayed in Figs. 11a and 11b as purple open markers. The greedy algorithm performs excellently for both the fine and coarse test cases: it is able to identify sensor placements that produce *nDOFS* values that are greater than the corresponding values for equidistantly-placed sensors and that are similar to the upper bounds of Monte-Carlo-simulated configurations, regardless of the given $N_\theta$ value. Indeed for the coarse-mode test case (Fig. 11b), the greedy-simulated *nDOFS* values are even greater than the upper bound of the Monte-Carlo-simulated values. We speculate that this is because the number of Monte Carlo tries (1000) was not sufficient to generate a random sensor configuration matching the optimal configuration.

The question then arises, how do the optimal sensor locations determined for the fine-mode aerosol test case compare with those determined for the coarse-mode case? Figures 12a and b displays these optimal sensor locations for $N_\theta = 11$ (given the same basic, 1$\lambda$, *PF* only instrument configuration simulated in Fig. 11). In the fine-mode case (Fig. 12a), the 11 angular measurements/sensors are distributed relatively evenly across the full scattering angle range. However for the coarse-mode case, the sensors are concentrated in the forward and backward scattering directions. These differences are also reflected in Figs. 11a and b, which show that the differences between the greedy-simulated *nDOFS* values and the equidistant *nDOFS*





values are much smaller for the fine than the coarse test case. The different optimal sensor locations can be explained by the differences in the scattering phase functions of the respective test cases: specifically, the scattering phase functions of coarse mode aerosol particles are more forward focused and less-isotropic than those of fine-mode particles. This finding is consistent

with the result presented in Section 4.6.1 that the retrievals of coarse-mode aerosol properties are more sensitive to extreme angle truncation than retrievals of fine-mode properties.

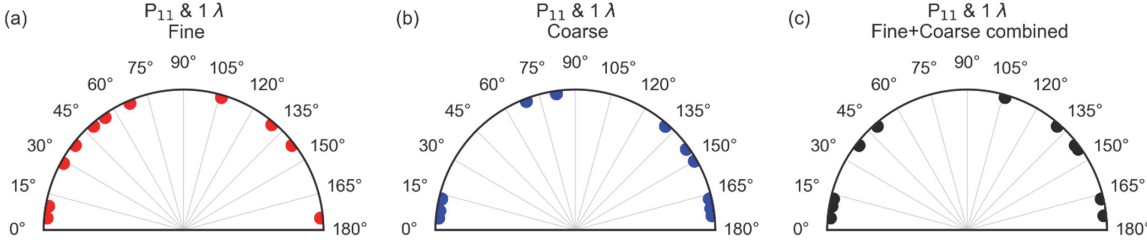

**Figure 12. (a) and (b) are optimal angular sensor locations for fine and coarse aerosol test cases, respectively, with phase function**
**only and phase function and polarized phase function measurement cases. (c) is the angular configuration constructed by combing fine and coarse configurations with $q = 6$ configurations, which due to a common angle point the resulting configuration has $N_{comb}$ = 11.**

Given the differences in the optimal sensor locations for the fine- and coarse-mode aerosol test cases, it can be expected that
using the optimally determined placements for one aerosol test case will produce sub-optimal results when applied to the other aerosol test case. Figs. 11a and 11b indicate that this is indeed the case. The optimal coarse-mode sensor placements produce $nDOFS$ values that are substantially lower than the optimal values when applied to the fine-mode aerosol test case (Fig. 11a), and vice versa (Fig. 11b). In both cases, even the equidistantly-placed sensors configuration yields higher $nDOFS$ values than using the optimal sensor placement of the wrong aerosol test case.


To mitigate this problem, one might attempt to combine the optimal sensor placements for both cases to create an instrument that has similarly high information content with respect to the measurement of both fine- and coarse-mode particles. To investigate this approach we created combined optimal configurations at each value of $N_\theta$ that were composed of optimal fine and coarse configurations. For example, to create the combined optimal configuration with $N_\theta = 11$, the optimal fine and coarse
configurations with $N_\theta = 6$ were joined together (since there was a commonly selected angle in both of these configurations the total number of unique sensors was only 11). The optimal sensor locations for this combined configuration are displayed are displayed in Fig. 12c. This examples shows that the combination method produces configurations that are less isotropic than the corresponding optimal fine-mode configuration, but not as forward-focused as the corresponding optimal coarse-mode





configuration. Figs. 11c and d plot the *nDOFS* values that are yielded from the optimal combined configurations as black open markers for the fine- and coarse-mode test cases, respectively. Generally, it is seen that the combined configurations produce *nDOFS* values that are only slightly sub-optimal, except for coarse-mode aerosol with $N_\theta < 10$ where the information content is substantially below optimum. This suggests that combining optimal configurations for different aerosol test cases is a promising approach for designing instrument configurations that are suitable for measuring a broader range of different aerosol types.

In this section we have used a case study to demonstrate that *DOFS* is a useful metric for optimizing the angular sensor placement in a polar nephelometer. Furthermore, we have shown that the optimal placements derived for the measurement of one type of aerosol will not necessarily coincide with the optimal placements determined for another aerosol type. As a final point, it should be stressed that these findings are only relevant only at low values of $N_\theta$, i.e., for fixed-detector systems. When the number of angular measurements is large, as in a laser imaging nephelometer, then there is no need to optimize the specific locations of those measurements.

## 5 Conclusion

Polar nephelometer designs can vary greatly in terms of spectral and polarimetric characteristics, as well as the number of angular measurements, their possible truncation, and their specific locations. These variations affect how much information can be retrieved from polar nephelometer measurements about the value of specific aerosol properties. To quantify these effects, we conducted a Bayesian sensitivity analysis to calculate an information content metric, *DOFS*, for a range of polar nephelometer instrument configurations, target aerosol cases, and assumed levels of measurement uncertainty and a priori knowledge. The majority of our analysis was focused on simulating the measurement of unimodal spherical aerosols described by three size distribution parameters (*VMR, GSD, V*) and $2N_\lambda$ refractive index parameters $n(\lambda)$ and $k(\lambda)$ (where $N_\lambda$ is the number of measurement wavelengths of the simulated polar nephelometer design). Given prior knowledge of the ranges of variability of these parameters in the atmosphere (Espinosa et al., 2019), the information content analysis yields $DOFS_i$ values near 1 for all of the parameters, even for a single wavelength polar nephelometer with no polarimetric capability. This suggests that even very basic polar nephelometers will provide useful, retrievable information when used for atmospheric measurements.

To assess benefit of polarimetric measurements for experiments with high prior knowledge on aerosol state parameters, and to ensure consistent comparison of the size distribution parameters across the fine and coarse cases, we additionally employed a percent-based a priori covariance matrix which was more stringent than the corresponding atmospheric-based a priori matrix. This led to a noticeable reduction of $DOFS_i$ over most state parameters and configurations. Consequently, the percent-based a priori method facilitated distinguishing between the information content of different instrument configurations. With the percent-based a priori method, $DOFS_i$ always increases as the measurement configuration becomes more comprehensive,





either through the addition of *PPF* and/or through the addition of multiple measurement wavelengths. However, in some cases the observed $DOFS_i$ increases are negligible, suggesting poor cost to benefit ratio for such added measurements. Adding extra measurement wavelengths to a 1λ configuration can significantly improve the information content for size distribution parameters (mean radius, GSD, and volume concentration). However, the addition of an IR wavelength to the 3λ setup (i.e., to

make a 4λ instrument) proved to be less beneficial in increasing $DOFS_i$. Furthermore, the *PPF* measurements appeared to be generally less informative using measurement error assumptions derived from an existing in situ instrument (Dolgos and Martins, 2014). However if the noise level is sufficiently low, *PPF* measurements can significantly improve the information content over all the state parameters. For some parameters, this even results in $DOFS_i$ values for a 1λ instrument with *PPF* measurement capability that are similar to the corresponding values for a 3λ or 4λ instrument with *PF* measurement only.


Comparing absorbing vs non-absorbing unimodal aerosol test cases revealed the unique nature of the state parameter *k*. In particular, $DOFS_k$ values are systematically larger for non-absorbing than absorbing aerosol test cases, and for coarse rather than fine aerosol test cases. We demonstrated that this is because the *PF*'s of non-absorbing/coarse particles are more sensitive to perturbations in *k* than the *PF*'s of absorbing particles Similar behaviour was not observed for any of the other state

parameters.

By considering a more complex bimodal non-spherical aerosol model, we showed that conducting more comprehensive spectral and *PPF* measurements can substantially improve the information content of different state parameters, such as *n* and *Sph%*.


We investigated the dependence of information content on angular truncation. For truncation in extreme forward and backward direction, $DOFS_i$ values generally decrease as the truncation angle increases, particularly for the coarse aerosol test cases. In contrast to the extreme truncation, the side truncation appears to have no significant impact on $DOFS_i$ reduction in both fine and coarse test cases. To investigate the dependence of information content on the number of angular measurements $N_\theta$ we

varied this parameter from 2 to 171. $DOFS_i$ increased with increasing $N_\theta$ over all the state parameters, but began to plateau out at $N_\theta$ values of around 20 – 40. This suggests that for the specific investigated cases, improving the angular resolution above a certain point does not provide substantial information gain. In addition, for some of the state parameters, the addition of multi-wavelength and *PPF* measurements at low $N_\theta$ values can lead to larger $DOFS_i$ increases than are possible purely through increase of $N_\theta$.


Finally, as a proof of concept *nDOFS* was employed as a metric for optimizing angular sensor placement using a greedy algorithm. It was demonstrated that for a given aerosol test case and $N_\theta$, the optimization algorithms finds sensor placements with greater *nDOFS* values than the corresponding equidistantly-placed sensors. Furthermore, combining the optimal sensor

placements from different aerosol test cases can be a viable approach for designing instruments that are suited to a broader
range of different target aerosols.

The results from this study provided insights on how different components involved in a Bayesian-based information content
analysis, such as the a priori covariance and aerosol model, could affect the outcome and interpretability of the data. Moreover,
the results from this study can help guide the future polar nephelometer designs and improve existing prototypes. Potential

follow-up studies could further expand the analysis to include more complex aerosol models (e.g. binned size distributions),
and to use more advanced forward models (e.g. models based on the discrete dipole approximation) to simulate non-spherical
aerosols such as soot.

**Data availability**

The original contributions presented in this study are included in the article and attached supplementary information. The

GRASP-OPEN model used to perform forward calculations is publicly available on the official GRASP website
(https://www.grasp-open.com/; last access: 27 May, 2022). The specific GRASP-OPEN outputs used in the study will be made
publicly available on Zenodo if the manuscript is accepted for publication.

**Acknowledgements**

The authors acknowledge funding from MeteoSwiss through a science project in the framework of the Swiss contribution to
the global atmosphere watch programme (GAW-CH) and from the Swiss National Science Foundation (BISAR project; SNSF
grant no. 200021_204823).

**Author contributions**

Conceptualization: Rob L. Modini, Martin Gysel-Beer, and Alireza Moallemi
Software: Alireza Moallemi, Tatyana Lapyonok, Anton Lopatin, David Fuertes, Oleg Dubovik,
Data curation: Alireza Moallemi
Funding acquisition and project administration: Martin Gysel-Beer
Supervision: Rob L. Modini, Martin Gysel-Beer
Writing - original draft: Alireza Moallemi, Rob L. Modini, and Martin Gysel-Beer
Writing – review & editing: Alireza Moallemi, Rob L. Modini, Tatyana Lapyonok, Anton Lopatin, David Fuertes, Oleg
Dubovik, Philippe Giaccari, Martin Gysel-Beer



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
