# Peer review of "Information content and aerosol property retrieval potential for different types of in situ polar nephelometer data"

_Atmospheric Measurement Techniques, 2022_

## Author Comment (AC1)

Dear Editor Brock and Reviewers Espinosa and Ahern,

Thank you for carefully considering our manuscript and for the positive and constructive comments. Please see below our point-by-point responses to these comments as well as the modifications that were made to the manuscript. The original comments are in black text, our responses are in blue text, and modifications to the manuscript are in **"bold blue text within quotation marks"**.

During the revision process, we discovered a mistake in Fig. 12 that required minor modifications. These modifications are also explained in detail below. They do not affect the main results and conclusions of the paper.

Best regards from the authors

**Responses to the Reviewers**

**Reviewer 1: Reed Espinosa**

**– General Comments –**
The manuscript describes a polar nephelometer information content study performed in the context of multiple polydisperse laboratory and ambient aerosols. The potential to retrieve aerosol concentration, size, spherical particle fraction and complex refractive index is evaluated for various instrument designs with different detection angles, wavelength configurations and polarization resolving capabilities. Overall, the work shows that retrievals of data from even a rudimentary polar nephelometer can theoretically provide a very significant amount of information on the sampled aerosol.

The content of the manuscript is novel, and it has the potential to be a very useful, and much needed, resource for future developers of polar nephelometer instruments. The material is inevitably a bit dense, but the text is well written, and the methods are clearly described. In my view, the manuscript could be slightly improved by the addition of a few more readily applicable results that would be more accessible to the casual reader. For example, it would be very informative to provide expected retrieval errors for the considered variables of state given the instrument configurations show in Figure 12 and the atmospheric measurement-derived a priori values. Overall though, the manuscript is clearly very appropriate for AMT, and I can recommend publication once the minor points below have been addressed.

We thank the Reviewer for their careful review of our manuscript and for their helpful and constructive comments.

Regarding the expected retrieval errors, although we agree that this would likely be more intuitive for the casual reader, we also believe that these errors would be open to misinterpretation. In order to obtain reasonable values one would need to have accurate knowledge of the underlying measurement errors (including any covariance). Since this is hardly possible in a general sense, we think it is more prudent to refrain from reporting expected retrieval errors in this case. Instead, our aim in Section 4.8 is to present a proof of concept study based on the *DOFS* metric that may later be adapted to more specific instruments and applications.

**– Specific Comments –**

1. Ln 71: Phase function is sometimes normalized but I believe absolute phase function (i.e., βsca*P11, where P11 is the phase function normalized such that the integral over all angles is 4π) is used here. It would be good to expectedly state the definition and/or units used for phase function in this work.

Thanks for the suggestion this is indeed the case. We have added this information to section 3.5, which now states:

**"The simulated *PF(θ, λ)* functions correspond to absolute phase functions with units of Mm$^{-1}$: they are normalized such that the integral over the solid angle equates to 4π$β_{scat}$, where $β_{scat}$ is the integrated scattering coefficient."**

2. Eq 2: I believe this equation is only valid for systems in which F(x) is linear everywhere, not just locally. Since Mie (and spheroid scattering) is very non-linear, I'm wondering if it is appropriate here.

Our interpretation of Eq. 2 is based on the cited reference Rodgers (2000), as highlighted in Section 2. I.e., we consider that the linearization around a local reference point (i.e., the kernel matrix **K** is a function of $x_0$) can be assumed when the forward model, F(x), is "…sufficiently linear within the error bounds of the retrieval." We believe this is a reasonable assumption for the cases we have considered and the Mie forward model. We also note that this assumption is consistent with the numerous previous studies cited in the introduction that have applied the same theoretical framework and Mie theory to aerosol remote sensing problems.

The implication of this approach (i.e., the dependence of **K** on $x_0$) is that one must simulate different aerosol test cases. We thought it was worth to add the following statement to the beginning of Section 3 to make this explicit, especially for readers that are not familiar with the theory:

**"Non-linearity of light scattering as a function of aerosol state parameters is a central aspect of the inverse problem of aerosol polarimetry. One effect is that the information content of a polarimetric measurement also depends on the properties of the aerosol under investigation (besides dependence on instrument features and a priori knowledge on the properties of the aerosol sample). Therefore, our general approach is to investigate different aerosol test cases, e.g. fine versus coarse mode aerosol, or non-absorbing versus absorbing aerosol."**

3. Sec 3.1.3: Is the angular Field-Of-View (FOV) of the sensor at a given scattering angle assumed to be negligibly small? This is not always the case for real instruments and should be clarified in the text. On a side note, in the PI-Neph Δθ≈0.2° but, due to smearing of the image by imperfect optics, the FOV of each pixel ends up being about ~1°, with significant overlap between pixels. Ultimately, data is reported at 1° resolution to avoid the need for a complicated deconvolution procedure.

This is an excellent point, which was also raised by Reviewer #2. Indeed, we have assumed that the angular FOV of each sensor is infinitesimally small. It is interesting to know those details for the PI-Neph and we agree that it is important to state that this assumption is not always valid for real instruments. To do so we have added the following statement to the end of section 3.1.3:

**"In all cases we assume that the angular field-of-view of each sensor is infinitesimally small (i.e., that there is negligible overlap between adjacent sensors). This simplification has little effect on resulting information content as long as the angular field-of-view of the sensors is small (e.g. ~1°) and less than the angular separation between adjacent sensors, whereas the results do not apply if the sensors have a wide field-of-view. In laser-imaging type nephelometers, the maximal number of informative angular measurements is typically limited by the effective field of view of the pixels rather than the angular separation between adjacent pixels."**

4. Ln 292: Covariance in polar nephelometer error can be large and quite complex, especially in imaging nephelometers. Do the authors have a sense of how sensitive their results are to this assumed covariance? How was the value $\rho=0.7$ selected?

The value of 0.7 was chosen to represent the case of substantial correlation between sensors up to an angular difference of around 10°. Specifically, $\rho = 0.7$ leads to correlation coefficients of ~0.7 between adjacent angular measurements, and correlation coefficients < 0.05 for measurements separated by more than 10°. We did not assess the sensitivity of the results to $\rho$. Such a sensitivity analysis was performed by one of the original studies that presented this approach (Knobelspiesse et al., 2012). Based on that sensitivity study we do not expect substantial sensitivity of our results to the assumed covariance, certainly in terms of the relative order of the information content results.

5. Table 3: This table defines the PSD using GSD while the following two tables use ln(GSD). It might be clearer to use a consistent metric throughout the manuscript.

Thanks for the suggestion. We changed the values in Table 3 from GSD to ln(GSD) to maintain consistency.

6. Ln 432: It would be good to provide a reference supporting the idea that refractive index values have significant spectral correlation. Two possible candidates would be Xu et al. (2019) and Gao et al. (2018).

Thanks for the suggestions; we have added these two references.

7. Table 5: I'm having trouble tracking which refractive indices were used to determine the percentage-based a priori covariance values in the bottom row. Each cell of the bottom row has two values: an absolute quantity and a percentage. My understanding from Figure 5 is that the absolute quantity listed is actually used for all $\lambda$ and aerosol species in the information content calculations. If so, which wavelength and species does the percentage shown apply to? Please clarify.

Thanks for picking this up and we apologize for the confusion. Indeed, as suggested by the previous Fig. 5, absolute a priori uncertainties were used for the refractive index parameters. This was incorrectly communicated in Section 3.4 and Table 5 in the original submission. This mistake also leads us to believe that the overall discussion of the percentage-based a priori ranges was generally too confusing and prone to error. Therefore, we have made the following changes to both rectify the communication error and to simplify the overall discussion:

- Rather than use separate percentage values $P_i$ for different variables, we now use a constant value of 3% for all of the size distribution related parameters. This simplification pushes some of the associated $DOFS_i$ values further away from 0.5 (e.g. see the revised Fig. 5) but extreme values close to 0 or 1 are still avoided, so comparisons between the different instrument configurations are still possible.
- Percentage-based a priori ranges are not used for the refractive index parameters since it is not possible to choose a common percentage value for the $\sigma_a$ for $k$ that is meaningful for both the non-absorbing (DEHS) and absorbing (BrC) aerosol test cases. Instead, we use constant absolute values for these $\sigma_a$ parameters. We have updated the absolute $\sigma_a$ values to 0.01 and 0.001 for $n$ and $k$, respectively. These updated values are chosen to maintain an effective comparison of the fine- and coarse-mode aerosol test cases (as previously), but also to provide some insight into the level of precision that might be expected in $n$ and $k$ retrievals, as suggested by the Reviewers comment below. E.g., $DOFS_i$ values larger than 0.5 now indicate that a particular measurement is informative enough to provide useful information at the 2nd and 3rd decimal places for retrieved $n$ and $k$, respectively.
- We have renamed the 'percentage-based' a priori selection method to the 'high level of prior knowledge' selection method.
- In the original submission, Table 5 reported the a priori values as ranges (i.e. $2\sigma_a$), while Fig. 5 reported them as uncertainties (i.e., $\sigma_a$). The values in Table 5 have been changed to uncertainties to ensure consistency in communication.

These updates led to the following changes in the revised manuscript:

- Section 3.4 and Table 5 have been updated to clarify that percentage-based values were only chosen for the size distribution parameters, while absolute values of 0.01 and 0.001 were chosen for $n$ and $k$, respectively. The reasons given above for choosing these absolute values have also been added to the revised Sections 3.4 and 4.2.
- Fig. 6 in the original submission has been removed from the revised manuscript. This figure displayed selected values from Figs. 5 and S5, and was originally used to highlight the substantial differences in the $DOFS_k$ values for the non-absorbing and absorbing aerosol test cases. We believe it no longer makes sense to highlight this difference with a stand-alone figure, since the difference is mainly because the constant a priori uncertainty of 0.001 is a much greater relative fraction of the reference $k$ for the non-absorbing case (1000%) than the absorbing case (0.93%). Section 4.4 has also been written to reflect this change, with less focus on the absorbing vs. non-absorbing comparison and more focus on the fine vs. coarse comparison. The revised Section 4.4 also takes into account the points raised by the Reviewers in the comment on $k$ below. The complete Section 4.4 is copied below this list.
- All instances of 'percentage-based' a priori selection method have been changed to 'high level of prior knowledge' selection method.

- Table 5 now displays the chosen a priori values as $\sigma_a$ values (rather than ranges $2\sigma_a$), to ensure consistency with how they are displayed in Fig. 5.
- The new $\sigma_a$ values meant that it was necessary to re-run the *DOFS* calculations and update Figs. 5, 6, 8, 9, 10, 11, 12, S2, S3, S4 and S5 in the original submission (Fig. 6 has been removed from the revised submission as explained above). While this caused slight changes in the absolute *DOFS* results, the overall trends are not affected, and the main results and conclusions of the analysis remain unchanged. This can be seen by comparing the original and revised figures in the track-changed version of the revised manuscript. Where appropriate, absolute *DOFS* values reported in the main text have been updated based on the new results.

The revised Section 4.4:

[revised manuscript text omitted]

8. Figure 4: Would it be possible to show the σ_a values within each subplot as they are in shown in Figure 5. This would greatly ease contextualization of these results for the reader.

Yes, we have added the $\sigma_a$ values to Fig. 4. As described in the response above, we have also updated Table 5 so that it displays $\sigma_a$ values rather than ranges (i.e. $2\sigma_a$), which we believe will further aid the reader by making a better link between the methodology and results sections.

9. Figure 5: In the coarse, k subplot, I interpret σ_a=0.0005 and DOFS≈1 for even the single wavelength non-polarized nephelometer to mean that the corresponding instrument has the potential to retrieve k to an accuracy much better than 0.0005. Although coarse mode state may have been slightly different, prior work has not noted very significant changes in PF resulting from changes in k as small as Δk=0.0005 (e.g., see Figure 2 of Espinosa et al. (2019)). Section 4.4 and Figure S4 provide a bit more context regarding the situations in which the present authors have observed PF to change significantly with k but I'm wondering if any intuitive explanation of the exact mechanism driving this high sensitivity is available, given that this feature has not been observed in prior work.

We appreciate this very insightful comment both for its interpretation of $\sigma_a$ and the link to the previous study of Espinoza et al., (2019). We have considerably rewritten the discussion of retrieving $k$ (Sect. 4.4), as detailed in our response above. The following excerpts are relevant in the context of this comment:

**"The greater $DOFS_k$ values for coarse compared to fine aerosol is consistent with previous phase function sensitivity calculations (see Figs. 1 and 2 in Espinoza et al., 2019)."**

**"We suggest that the greater amount of internal light absorption within coarse versus fine particles has a stronger feedback on internal light intensity and hence greater perturbation of the external scattered light field for equal change in $k$."**

**"$DOFS_k$ values noticeably greater than zero are obtained except for the fine mode absorbing example, which suggests that even under the assumption of very high prior knowledge on $k$, angularly-resolved light scattering measurements can still contribute additional information about aerosol absorption. For example, the $DOFS_k$ values for the non-absorbing, coarse aerosol case are unity, which suggests that for this simple unimodal aerosol and assumed measurement uncertainty, it should be possible to retrieve $k$ to a precision better than three decimal places (i.e., given that the a priori uncertainty $\sigma_{a,k}$ is 0.001)."**

**"The results presented here demonstrate that polarimetric measurements can be very informative on $k$ when probing simple aerosols. However, atmospheric aerosol samples are much more complex in terms of size dependent composition, mixing state and particle shape. This considerably reduces the information content of polarimentric measurements with respect to light absorption."**

We also believe it is worth highlighting this finding, so added the following statement to the Abstract: **"Nevertheless, we show that in this situation polar nephelometers can still provide informative measurements: e.g. it can be possible to retrieve the imaginary part of the refractive index with high accuracy, if the laboratory setting makes it possible to keep the probed aerosol sample simple."**

10. Figure 6: Are these DOFS (and the results in Fig S4) based on the percentage- or atmospheric-based a priori variance values?

They were based on the percentage-based a priori variance values (now referred to as the high level of prior knowledge method). However, as discussed in the comment above this figure has now been removed from the manuscript.

This comment made us realize that this information was not provided in a number of other places throughout the manuscript (e.g. Section 4.6 and the captions of Figs. 7, 8 and 9, which were Figs. 8, 9 and 10 in the original submission). We have now added the information in sentences such as: **"These results were obtained using default measurement noise values, and the "high level of prior knowledge" method for selecting a priori variances"**

11. Ln 750: Do the authors know of a particular nephelometer that suffers from side angle truncation? If so, it would be good to add a reference to this instrument. If no reference is available, it may be better to soften this statement and say that some designs could potentially suffer from side angle truncations.

We are currently working with a laser imaging nephelometer that suffers from side angle truncation. We modified Section 3.1.2 to make this clear: **"We currently test and validate a laser imaging nephelometer similar in design to the instrument by Dolgos and Martins (2014), which suffers from a gap in measurements near 90° scattering angle. Here, we refer to this type of measurement gap as side angle truncation…"**

The text in Section 4.6.2 now reads:
**"As explained in Section 3.1.2, some polar nephelometers suffer from side angle truncation…"**

12. Ln 777: Intuitively, I imagine there to be two relatively separate mechanisms that lead to improvements in DOFS with increasing N_θ: (1) an improved ability to capture angular features in the PF and PPF that encode information about the aerosol and (2) an increase in measurement statistics that helps to beat down noise and effectively increase the accuracy of the measurement. The two mechanisms are likely quite difficult to disentangle but I'm wondering if the authors have any sense of their relative contributions here. If mechanism (2) was dominate, I would expect the N_θ value where "plateauing" starts to occur to be strongly dependent on the assumed error covariance (specifically the value of ρ). Is the conclusion that the plateau generally occurs 20<N_θ<40 robust to different choices of ρ? This could be quite relevant in terms of instrument design considerations where there is frequently a choice between adding more angles or increasing the accuracy in a smaller subset of angles.

We agree that this is an interesting point and of potential practical importance in terms of a possible trade-off between number of detection angles and accuracy (e.g. potentially when comparing a fixed detector type nephelometer with a laser imaging nephelometer). However, to properly investigate this question would require a substantial new sensitivity analysis and additions to the manuscript, which we do not attempt. We have already taken care in Section 4.7 to highlight that the plateau region only applies to the specific combination we have investigated (i.e., simple aerosol model, high level of prior knowledge, measurement noise).

One important aspect is that more quantitatively interpretable results require a detailed error model for the instrument because error covariance is much more complex than considered with the "ρ-value" approach, and error covariance also depends on the aerosol test case. As a side note: We have developed such an error model for our own instrument. However, it is very difficult to validate the error model within tight margins given lack of suitable reference aerosols with accurately known phase function.

13. Ln 781: I would recommend restating the sentence that begins on this line. As it is currently written, it almost sounds like the plateau in IC is more prominent with complex particles or low instrument noise, which I think is the opposite of what the authors intended.

Thanks for the suggestion we have modified the sentence accordingly. It now states: **"Based on results in the previous sections, it can be expected that the plateau in information content as a function of $N_\theta$ shifts to higher $N_\theta$ values e.g. for more complex aerosol models or smaller measurement noise."**

14. Ln 784: It may be worth noting that these conclusions all apply only to polydisperse aerosols. Monodisperse aerosols, or even reactively narrow polydisperse size distributions, will have significantly more angular features and likely continue to benefit from more angles, well beyond the plateaus observed here.

More pronounced features in the polarized phase functions of narrower size distributions increases the information content for fixed number of angles and measurement error. Whether or not the plateau occurs at similar or different $N_\theta$ is more difficult to answer and we do not dare to speculate on it. Inclusion of such additional analyses is beyond the scope of this manuscript. To note this issue we have added the following statement to Section 4.7: **"Finally, it should be noted that these results only apply to polydisperse aerosol size distributions. Information content and its dependence on instrument design features differs considerably when probing narrow size distributions with more pronounced angular features in the polarised phase function."**

**– Technical Corrections –**
Ln 88: "Polarized Imaging Nephelometer" should be capitalized.
Implemented

Ln 290: "Wavelength" should be one word
Implemented

Ln 446: This sentence contains an extra "it".
Implemented

Ln 456: I might suggest something like "detection angles" in place of "sensor" since some instruments (e.g., Imaging Nephs) only have a single CCD sensor.
Implemented

Ln 471: Should read "...with an increasing..."
Implemented

**– References –**
Xu, Feng, et al. "A correlated multi-pixel inversion approach for aerosol remote sensing." Remote Sensing 11.7 (2019): 746.

Gao, M., Zhai, P.-W., Franz, B., Hu, Y., Knobelspiesse, K., Werdell, P. J., Ibrahim, A., Xu, F., and Cairns, B.: Retrieval of aerosol properties and water-leaving reflectance from multi-angular polarimetric measurements over coastal waters, Opt. Express, 26, 8968–8989, https://doi.org/10.1364/OE.26.008968, 2018.

Reviewer 2: Adam Ahern

The manuscript "Information content and aerosol property retrieval potential for different types of in situ polar nephelometer data" by A. Moallemi et al. presents an evaluation of the information content of data from different polar nephelometer configurations for a variety of simple and complex aerosol models. They present a Bayesian sensitivity analysis with an appropriate discussion of the impact of prior assumptions and are deliberate in communicating the limitations of the analysis.

They use the Degree of Freedom for Signal (DOFS) to quantitatively compare polar nephelometer designs while varying amounts of signal truncation, number and position of detectors, and number of investigated wavelengths. This work represents a valuable contribution to the field because, similar to the work of Knobelspiesse et al. for remote sensing instruments, Moallemi et al. quantitatively explore the connection between in situ instrument design and the retrieved parameters. This was well-illustrated by the use of DOFS and the reductive greedy algorithm to optimize detector placement.

This manuscript is excellent and I could recommend it for publication in its current state, although I will use this opportunity to make a few small comments.

General comment:

Although the manuscript is impressive in the scope of the design permutations explored, I think that the fundamental choice of which wavelength(s), as opposed to how many wavelengths, to investigate is taken for granted. This might make an interesting addition to the supplemental material.

We thank the Reviewer for their careful review of our manuscript and for their helpful and constructive comments.

We agree that the choice of which wavelength(s) is an interesting question. However, the range of the investigated design choices is already very large, as mentioned by the Reviewer. Further investigation of particular wavelengths would expand the range of investigation dramatically. Therefore, we think such an investigation is beyond the scope of the manuscript. Nevertheless, we think it is a good point and have added it to the conclusion as a possible follow up study: **"Potential follow-up studies could further expand the analysis to include more complex aerosol models (e.g. binned size distributions), to investigate and compare specific measurement wavelengths (i.e., by varying the chosen measurement wavelengths, rather than simply their number), and…"**

Minor comments:

3.1.3 Angular characteristics: number of proved angles assuming evenly distributed measurements

P10.148 Is it true that each data point represents a theoretical "sensor" that is infinitely narrow? As opposed to a sensor that has a non-zero solid angle?

This is an excellent point, which was also raised by Reviewer #1. Yes, we have assumed that the angular FOV of each sensor is infinitesimally small. To clarify this we have added the following statement to the end of section 3.1.3:

**"In all cases we assume that the angular field-of-view of each sensor is infinitesimally small (i.e., that there is negligible overlap between adjacent sensors). This simplification has little effect on resulting information content as long as the angular field-of-view of the sensors is small (e.g. ~1°) and less than the angular separation between adjacent sensors, whereas the results do not apply if the sensors have a wide field-of-view. In laser-imaging type nephelometers, the maximal number of informative angular measurements is typically limited by the effective field of view of the pixels rather than the angular separation between adjacent pixels."**

3.5 Forward Model

P17.443 Besides Espinosa et al., consider including Schuster et al. (2019)

Thanks for the suggestion we have added this reference.

4.1 Dependence of information content on the angular configurations of previous polar nephelometer designs

P18.483 The way this is discussed is a little confusing because nDOFS is an analytical result. I wonder if another way to discuss this is that the nDOFS presented is specific to your test aerosol parameters. To extrapolate more broadly, i.e. if you want to compare which instrument provides more information about fine aerosol parameters (of which your test aerosol is a subset), then you must consider the sensitivity of nDOFS to the aerosol model parameters in the range of interest.

We have modified the sentence to better highlight that the results in Fig. 3 are specific to this specific aerosol test case, as suggested. The sentence now reads: **"Instead, the four sensor instrument may be optimized for probing aerosols similar to this specific test aerosol case (unimodal, non-absorbing, fine mode aerosol), whereas the seven sensor instrument may have be optimized for different target aerosol properties."**

Fig. 5. Consistency of labels with Fig. 4 would be nice (e.g. VMR vs Median Radius)

Implemented, thanks for picking this up.

4.4 Information content for the imaginary part of the refractive index

P26.674 where **the** latter is equivalent

Implemented

P27.684 **atmospheric**-based a prior

Implemented

4.8 Proof of concept for using DOFS as metric for optimizing angular sensor placement

Fig. 11 Consider using different marker shapes. The blue and black are hard to differentiate.

Implemented: the black open circles have been changed to black stars.

P33.843 Labels for Fig. 11 state PF and PPF, whereas this line states only PF.

Thanks for picking this up. The text has been changed from "PF-only" to **"PF and PPF"**. This comment also led us to discover a mistake in the original submission as explained below in the Section 'Other changes to the manuscript'.

5 Conclusion

P35.900 To assess **the** benefit

Implemented

Dick, W. D., Ziemann, P. J., and McMurry, P. H.: Multiangle Light-Scattering Measurements of Refractive Index of Submicron Atmospheric Particles, Aerosol Sci. Technol., 41, 549–569, https://doi.org/10.1080/02786820701272012, 2007.

Li, D., Chen, F., Zeng, N., Qiu, Z., He, H., He, Y., and Ma, H.: Study on polarization scattering applied in aerosol recognition in the air, Opt. Express, OE, 27, A581–A595, https://doi.org/10.1364/OE.27.00A581, 2019.

Nakagawa, M., Nakayama, T., Sasago, H., Ueda, S., Venables, D. S., and Matsumi, Y.: Design and characterization of a novel single-particle polar nephelometer, Aerosol Sci. Technol., 50, 392–404, https://doi.org/10.1080/02786826.2016.1155105, 2016.

Schuster, G. L., Espinosa, W. R., Ziemba, L. D., Beyersdorf, A. J., Rocha-Lima, A., Anderson, B. E., Martins, J. V., Dubovik, O., Ducos, F., Fuertes, D., Lapyonok, T., Shook, M., Derimian, Y., and Moore, R. H.: A Laboratory Experiment for the Statistical Evaluation of Aerosol Retrieval (STEAR) Algorithms, Remote Sens., 11, 498, https://doi.org/10.3390/rs11050498, 2019.

**Other changes to the manuscript**

- When responding to a minor comment from Reviewer #2 we discovered that an incorrect version of Fig. 12 (now Fig. 11 in the revised submission) had been copied into the manuscript. The incorrect figure showed the optimal angular configuration for a PF-only instrument, rather than for an instrument with PF and PPF capabilities, as

was intended. The correct figure corresponding to an instrument with PF and PPF capability has now been added to the revised manuscript. The results displayed in the updated figure are entirely consistent with those in the previous figure, and therefore also with the explanations and discussions already provided in Section 4.8. Only some minor details have changed (e.g. the combined fine and coarse configuration now has 10 unique angles as opposed to the 11 it had previously). These minor details have no influence on the main results and conclusions of the Section.

- A typo in Section 2 was rectified (Jacobin changed to Jacobian). Thanks Editor Brock for picking this up.